The cranial endocast of the Upper Devonian dipnoan ‘Chirodipterus’ australis

Henderson Struan A.C. S.Henderson-16@sms.ed.ac.uk struanhenderson@gmail.com
Challands Tom J.
School of Geosciences, University of Edinburgh , Edinburgh , United Kingdom
Knoll Fabien
Electronic publication date: 2018 Jul 6
Publication date: 2018
Volume: 6
Electronic Location ID: e5148
Received 2018 Feb 28; Accepted 2018 Jun 12
Copyright: ©2018 Henderson and Challands
Copyright year: 2018
Copyright holder: Henderson and Challands
License: This is an open access article distributed under the terms of the Creative Commons Attribution License, which permits unrestricted use, distribution, reproduction and adaptation in any medium and for any purpose provided that it is properly attributed. For attribution, the original author(s), title, publication source (PeerJ) and either DOI or URL of the article must be cited.
License URL: https://creativecommons.org/licenses/by/4.0/

Keywords: Endocast, Dipnoi, Devonian, Sarcopterygii, Chirodipterus, Inner ear

Funding: Callidus Services Ltd Struan Henderson received no funding for this work. Dr Tom Challands is supported by Callidus Services Ltd. The funders had no role in study design, data collection and analysis, decision to publish, or preparation of the manuscript.

==============================
One of the first endocasts of a dipnoan (lungfish) to be realised was that of the Upper Devonian taxon Chirodipterus australis. This early interpretation was based on observations of the shape of the cranial cavity alone and was not based on a natural cast or ‘steinkern’ nor from serial sectioning. The validity of this reconstruction is therefore questionable and continued reference to and use of this interpretation in analyses of sarcopterygian cranial evolution runs the risk of propagation of error. Here we present a new detailed anatomical description of the endocast of ‘Chirodipterus’ australis from the Upper Devonian Gogo Formation of Western Australia, known for exceptional 3D preservation which enables fine-scale scrutiny of endocranial anatomy. We show that it exhibits a suite of characters more typical of Lower and Middle Devonian dipnoan taxa. Notably, the small utricular recess is unexpected for a taxon of this age, whereas the ventral expansion of the telencephalon is more typical of more derived taxa. The presence of such ’primitive’ characters in ‘C.’ australis supports its relatively basal position as demonstrated in the most recent phylogenies of Devonian Dipnoi.

Introduction

A monophyletic group of sarcopterygians, the Dipnoi originated in the Early Devonian around 400 Ma, and are still extant today (Denison, 1968; Thomson & Campbell, 1971; Campbell & Barwick, 1982; Campbell & Barwick, 2000; Chang, 1984; Schultze, 2001; Qiao & Zhu, 2015). The extensive lungfish fossil record during their rapid diversification throughout the Devonian Period allows an unprecedented opportunity to examine a discrete evolutionary trajectory of neurocranial development early within a clade’s history. Furthermore, examination of lungfish brain morphology indirectly from cranial endocasts of the brain cavity provides important information concerning overall cranial disparity of the Sarcopterygii as a group concomitant with the first appearance of tetrapods and tetrapodomorphs (Cloutier & Ahlberg, 1996; Zhu & Yu, 2002; Friedman, 2007).

‘Dipnoan’ was one of the eight basic brain plans of extinct and extant vertebrates proposed by Stensiö (1963), the remaining seven being osteolepiform, porolepiform, actinistian, actinopterygian, elasmobranchiomorph, petromyozontid and myxinoid. The dipnoan endocranium was described as platybasic, where the cavity for the brain projects between the orbits (which are separated) and the endocranium has a broad base, though Stensiö went on to detail definitive neural characters of the ‘dipnoan’ brain type: (1) overall shape long and narrow; (2) closely separated narrow, deep hemispheres; (3) high cerebral hemispheres extending anteriorly; (4) inverted or slightly inverted telencephalon; (5) anterodorsally situated olfactory bulbs at, or just in front of, the cerebral hemispheres in adult forms; (6) large hypothalamus; (7) reduction or disappearance of the lobus posterior hypothalami; (8) slight development or absence of saccus vasculosus; (9) hypophysis situated posterior to the fundibulum; (10) a narrow mesencephalon; (11) a small metencephalon; and (12) a long myelencephalon. It is unlikely, given the lack of brain histological data from fossils, that these defining characters can easily be applied to endocasts of fossil lungfish. Stensiö’s inferences were largely based on the brains of the extant African (Protopterus annectens) and Australian (Neoceratodus forsteri) lungfish, though he also drew comparisons with the Upper Devonian Chirodipterus wildungensis (from Säve-Söderbergh, 1952). His conclusion that no major changes in gross dipnoan brain morphology had occurred since the Devonian has been challenged in the light of new studies on Early, Middle and Late Devonian lungfish (Campbell & Barwick, 1982; Campbell & Barwick, 2000; Clement & Ahlberg, 2014; Challands, 2015; Clement et al., 2016), and is challenged further herein.

Comparison of primitive lungfish neural characters with those shared by more basal sarcopterygians (e.g., Youngolepis) and tetrapodomorphs allows the polarity of endocranial characters to be ascertained and necessitates the examination of homologous structures in more basal outgroup taxa. For example, the neurocranium of the early Pragian tetrapodomorph Tungsenia paradoxa features similar olfactory tract angle, separate pineal and parapineal organs and similar nerve V positioning to the basal dipnomorphs Youngolepis, Powichthys, Porolepis and Glyptolepis (Lu et al., 2012) and so these conditions are regarded as primitive for these taxa.

The neural characters of the oldest lungfish, Diabolepis speratus, from Yunnan in South China (Lochkovian Age) have not currently been determined (Chang, 1984; Smith & Chang, 1990) and so the character states for the most primitive lungfish remain unknown. The primitive condition of neural features for the Dipnoi were therefore suggested by Campbell & Barwick (2000) on examination of the internal and external neurocranial features of Dipnorhynchus kurikae, a basal Early Devonian lungfish, and through comparison with the brain of N. forsteri. These features were: (1) no ventral expansion of the telencephalon (brain region associated with olfaction and vision); (2) posterior position of nerve II; (3) small utricular recesses; (4) large ampulla on the posterior semicircular canal (relative to younger Devonian Dipnoi). They further suggest that olfactory tracts with pedunculate bulbs (as opposed to sessile bulbs sitting in the forebrain) and narrower semicircular canals are primitive. This model has recently been, for the best part, validated by the recent digital endocast of Dipnorhynchus sussmilchi (Clement et al., 2016), though sessile rather than pedunculate olfactory bulbs were observed contradicting the hypothesis of Campbell & Barwick (2000). Other endocranial characters were also highlighted as potentially primitive, notably (1) the absence of a separate sacculus and lagena; (2) a sinus superior that does not extend dorsally above the rhombencephalon; (3) a high angle between the anterior and posterior semicircular canals; (4) the presence of median dorsal canals posterior to pineal—parapineal and, (5) bifurcation of the anterior cerebral vein within the cranial cavity.

An extensive  μCT cranial endocast of Dipterus valenciennesi, a Middle Devonian lungfish, revealed crucial anatomical features that realised polarity of phylogenetically informative endocast characters in the Dipnoi (Challands, 2015). These characters differ subtly from the predictions of Campbell & Barwick (2000) and include: (1) ventral expansion of the telencephalon seen in Rhinodipterus kimberleyensis (Clement & Ahlberg, 2014) and extant dipnoans (derived), but lacking in the basal Dipnorhynchus sussmilchi and Dipterus valenciennesi (primitive); (2) numerous dorsomedial canals and separate pineal and parapineal recesses (primitive), compared to the absence of dorsomedial canals and a shared pineal-parapineal recess (derived) found in Upper Devonian lungfish crownward of Dipnorhynchus sussmilchi (Dipterus valenciennesi is intermediate having one medial canal and a shared recess); (3) enlarged utricular recess (derived—extant lungfish have large utricular recesses relative to Dipterus valenciennesi and Rhinodipterus kimberleyensis, which have enlarged recesses relative to Dipnorhynchus kurikae); (3) size of the posterior semicircular canal ampullae—inflated is the primitive condition seen in Dipnorhynchus kurikae while small and deflated is the derived condition seen in younger lungfishes. Challands (2015) corroborates conclusions that younger Devonian lungfish (Rhinodipterus kimberleyensis (Clement & Ahlberg, 2014), Chirodipterus wildungensis (Säve-Söderbergh, 1952) and Griphognathus whitei (Miles, 1977)) had derived brains. In light of this new information, the current study provides a detailed anatomical description of the endocast of ‘Chirodipterus’ australis to further assess the polarity of neural characters, but also to ultimately contribute new information to determining the true relationships of this genus which is regarded as being polyphyletic (Friedman, 2007; Qiao & Zhu, 2009; Pardo, Huttenlocker & Small, 2014; Challands, 2015; Challands & Den Blaauwen, 2017).

‘Chirodipterus’ australis and the genus Chirodipterus

‘Chirodipterus’ australis Miles, 1977 is a Late Devonian (mid-Frasnian) short-snouted lungfish from Gogo, north Western Australia (Fig. 1). Several well preserved three dimensional specimens of ‘Chirodipterus’ australis have been collected from this locale and are ideal for detailed study using μCT scanning to reveal the endosseous cranial features.

Figure 1 Map of the Gogo Formation in the Kimberley region of Western Australia.

Map of the Gogo Formation in the Kimberley region of Western Australia, modified from Clement (2012). Numbers are locality numbers taken from Miles (1971) and represent the sites that the ‘Chirodipterus’ australis specimens were collected from (80-NHM PV P56035; 91-NHM PV P56038).

Miles (1977) assigned ‘C’. australis to Chirodipterus based on entopterygoid structure, conjoined denticles and possession of one subopercular bone. The phylogenetic position of ‘Chirodipterus’ australis and its belonging to the genus Chirodipterus is, however, debated (Friedman, 2007). Indeed, monophyly of Chirodipterus does not stand up to scrutiny and ‘Chirodipterus’ australis, alongside C. wildungensis, C. liangchengi, C. onawayensis and C. potteri, likely comprise a relatively derived, yet polyphyletic lungfish genus restricted to the Middle-Late Devonian. An extensive review of the morphology of four Gogo dipnoans (‘C’. australis, Gogodipterus paddyensis, Griphognathus whitei, Holodipterus gogoensis) highlighted some differences in the tooth-plates, nerves and vessels between ‘C’. australis and C. wildungensis (Miles, 1977), suggesting they belong to different genera—‘Chirodipterus’ australis dentition is undoubtedly unlike C. wildungensis and C. liangchengi (Kemp, 2001). Inclusion of recent character data in phylogenetic analyses places C. wildungensis as a derived Devonian lungfish (Challands, 2015). Further, C. liangchengi has been found to resolve in a polytomy between the most primitive and most derived Devonian Dipnoi, being recurrently placed basal to C. wildungensis (Friedman, 2007; Clement, 2012; Challands, 2015; Clement et al., 2016). The most recent published Devonian lungfish phylogeny (Clement et al., 2016) recovers either a monophyletic grouping of ‘Chirodipterus’ australis and Gogodipterus paddyensis within this polytomy, or places G. paddyensis basal to ‘C’. australis.

‘Chirodipterids’ (and ‘dipterids’) have been suggested to occupy a phylogenetic position between basal Devonian lungfish (e.g., Dipnorhynchus, Uranolophus) and derived Upper Devonian taxa (e.g., Griphognathus), with the proviso that Chirodipterus wildungensis is the only valid species of Chirodipterus (Friedman, 2007). The high quality of ‘Chirodipterus’ australis specimens, has caused further confusion by leading researchers to typically refer to material of ‘C’. australis when discussing the genus Chirodipterus, rather than the type species Chirodipterus wildungensis. If ‘C’. australis is a true member of the genus it would be predicted to possess a similar derived brain morphology to Chirodipterus wildungensis. Conversely, given its current phylogenetic position as being more basal, and polyphyletic, to C. wildungensis (Clement et al., 2016) the null hypothesis is that it comprises a melange of derived and primitive endocranial characters more akin to Dipterus valenciennesi. The description of the endocast of ‘Chirodipterus’ australis presented here provides a test of this hypothesis.

Materials and Methods

To account for intraspecific variation as best possible, two individual ‘Chirodipterus’ australis specimens, NHMUK PV P56035 and NHMUK PV P56038, were analysed (Fig. 2). These two specimens comprise complete and uncrushed crania previously prepared and exposed from the matrix by acid etching. NHMUK PV P56035 and NHMUK PV P56038 were two of twenty specimens studied by Miles (1977) for his extensive review on the cranial features of Gogo dipnoans, although neither specimen received particular examination in his study.

Figure 2 Photographs of specimens.

(A) NHMUK PV P56035 in right lateral aspect. (B) NHMUK PV P56038 in ventral aspect. Source credit: T. Challands.

Scanning and segmentation

NHMUK PV P56035 and NHMUK PV P56038 were scanned at the Natural History Museum, London, UK, using a Metris X-Tek HMX ST 225 CT system. The scanning parameters for NHMUK PV P56035 were: 210 kV; 200 µA; 2.5 mm copper filter; x-offset = −0.043; y-offset = −0.043; z-offset = 0.043; source to object = 253 mm; source to detector = 1,170 mm; and mask radius = 43.1652, 3,142 projections with an angular step of 0.115°. For NHMUK PV P56038, the scanning parameters were: 210 kV; 200 µA; 2.0 mm copper filter; x-offset = −0.049; y-offset = −0.049; z-offset = 0.049; source to object = 285 mm; source to detector = 1,170 mm; and mask radius = 48.7117, 3,142 projections with an angular step of 0.115°. The CTPro package with Beam hardening type = simple, was used to construct the projections. Voxel size was 43 µm for NHMUK PV P56035 and 49 µm for NHMUK PV P56038.

The raw sinogram μCT data was reconstructed as 32-bit greyscale tiff files, which were rotated and cropped in ImageJ (Schneider, Rasband & Eliceiri, 2012). The tiff image stack produced was then rendered and segmented using Materialise’s Interactive Medical Image Control System (MIMICS®) Research v18.0.0.525 (biomedical.materialise.com/mimics, Materialise NV, Leuven, Belgium), to produce a three dimensional (3D) endocast model. The endocasts were then also smoothed by a ‘smoothing factor’ of 0.2, with 5 iterations. Segmented images were also imported into the open-source software Drishti Volume Exploration and Presentation Tool v.2.0 (Limaye, 2012) to calculate the volumes of neural structures.

Measurements

Measurements were made on the endocasts of ‘Chirodipterus’ australis specimens NHMUK PV P56035 and NHMUK PV P56038. The measurements taken on the ossified labyrinth were as follows: angles between anterior-posterior, anterior-lateral and posterior-lateral semicircular canals (SCCs); internal circumferences of anterior, lateral and posterior SCCs; length of arc of lateral SCC; major axis of lateral SCC fenestra; minor axis of lateral SCC fenestra; radius of arc of lateral SCC; surface area of anterior, lateral and posterior SCC ampullae; volume of anterior, lateral and posterior SCC ampullae; surface area of utricular recess; volume of utriculuar recess; surface area of sacculolagenar; volume of sacculolagenar. Measurements are identical to those in Challands (2015, fig. 2; see also Fig. S2 for a guide on these measurements) on the endocast of Dipterus valenciennesi allowing direct comparisons of ‘Chirodipterus’ australis with Dipterus valenciennesi, yet also with Protopterus dolloi and Neoceratodus forsteri, for which measurements of the endosseous labyrinths were also reported. In addition to these measurements, ratios of the volume and surface area of the utriculus to that of the sacculus were calculated.

The following measurements were taken from the forebrain (telencephalon and diencephalon) of the specimens: length of olfactory nerve canals from the nasal sac to olfactory bulbs; length of the olfactory nerve canals from nasal sac to telencephalon; diameter of olfactory nerve canals (taken from mid-point of olfactory nerve canals); angle of direction of the olfactory nerve canals from the telencephalon; angle between olfactory nerve canals; length of olfactory bulbs; volume of the olfactory bulbs; volume of the ventral expansion of the telencephalon; width of nerve II; depth of nerve II. The ventral expansion of the telencephalon was defined (following Clement & Ahlberg, 2014, fig. 5) as the region of the endocast anterior and ventral to the ventral margin of the roots of the optic nerves and posterior and ventral to the ventral union of the olfactory bulbs and telencephalon. The olfactory bulbs were defined as from where the olfactory nerve canals begin to expand (dorso-ventrally and latero-medially) to the union with the main body of the forebrain.

In the hindbrain (rhombencephalon) measurements were taken for: width (medio-lateral direction) of the endolymphatic ducts; depth (antero-posterior direction) of the endolymphatic ducts.

The volumes of the telencephalon, diencephalon, mesencephalon and rhombencephalon were calculated with definitions of the regions as follows (see also, Clement & Ahlberg, 2014, fig. 5): telencephalon (excluding olfactory nerve canals)—region anterior to a line drawn from the root of nII to the anterior base of the pineal-parapineal recess, and posterior to the emanation of the olfactory bulbs; diencephalon—region between the posterior margin of telencephalon and a line drawn from the posterior base of the hypophyseal recess to the beginning of the dorsal expansion of the endocast; mesencephalon—region between the posterior margin of the diencephalon and the union of the main body of the endocast and the labyrinth; rhombencephalon—region posterior to the posterior margin of the mesencephalon (excluding the labyrinth); labyrinth—regions lateral to the rhombencephalon, specifically, lateral to the anterior division of the supraotic cavities. Results of all measurements are presented in Tables S1–S4.

Phylogenetic methods

The matrix of Clement et al. (2016) comprising characters derived from endocasts was used with the addition of ‘Chirodipterus’ australis. This matrix was analysed via parsimony analysis conducted using the implicit enumeration method in TNT V1.5 (Goloboff, Farris & Nixon, 2008). The implicit enumeration method finds an exact solution and all trees using branch and bound. Parsimony analysis was also conducted using stepwise addition with 10,000 random addition sequence replicates holding five trees at each step, with tree bisection and reconnection (TBR) enabled. Bremer indices were calculated using the bsupport command for trees suboptimal by 20 steps. The Late Devonian coelacanth Diplocercides was designated as the outgroup.

Preservation of specimens

The neurocranium of NHMUK PV P56035 is well preserved in 3D, with most of the cranial cavity preserved completely. Only small sections of the neurocranial walls are damaged, most noticeably in the hypophyseal recess and the skull roof anterior to the otic region. The neurocranium surrounding the hypophyseal recess has collapsed and is open to the parasphenoid, with no bone found anterior to the otic region and damage extending forwards to the mesencephalon. Matrix fills the majority of the cranial cavity and few pyrite concretions are found within the neurocranium. Acid preparation has been performed on this specimen, however where the acid has not reached the contrast of the μCT scan data is low.

NHMUK PV P56038 is less well preserved. The specimen has been compressed dorso-ventrally and has sustained considerable damage in the left postero-lateral region of the neurocranium such that the left labyrinth could not be segmented. A large crack runs parallel to the endocast through the region of the left labyrinth with the resultant space being filled with sediment. In this region, to the left and right of the midbrain, and in the rostral region surrounding the olfactory nerve canals, there are large concentrations of pyrite. These concretions are an impediment to the interpretation and segmentation. This specimen has also been acid prepared and contrast in the μCT scan data is low where the acid has not reached and the sediment has not been dissolved. Contrast is particularly low in the postero-ventral portion of the neurocranium and, as such, the notochord is not visible and cannot be segmented.

Results

Endocast anatomical description

The following description provides a detailed account of endocast morphology of ‘Chirodipterus’ australis, building upon interpretations made by Miles (1977). All features presented herein relate not to the true cranial nerves, vessels or brain, but to the endocast, however the endocast is regarded as being indicative of actual brain morphology in lungfish (Clement et al., 2015). The endocast of NHMUK PV P56035 is approximately 41 mm in length (from the bottom of olfactory nerve canals to the spinal cord) and 28 mm at its widest (across the semicircular canals), while NHMUK PV P56038 is 55 mm long and 30 mm wide. Both endocasts display slight asymmetry, which is known to occur in animals inhabiting stressed environments (Parsons, 1992), and is consistent with other specimens from the Gogo formation (Long & Trinajstic, 2010). It may be that this asymmetry is a reflection of the asymmetry in the dermal skull bones of Devonian lungfish, though the dermal bones of the specimens examined here could not be observed. This asymmetry is not an impediment to interpretation however. Volumes of the brain regions are presented in Table S1.

Nasal sac: The nasal sacs (nc, Figs. 3–8) are large and open ventrally, with a convex dorsal surface and no observable divisions. Miles (1977) states, however, that they are ovoid (see cav.nc, figs. 18 & 64, Miles, 1977), yet both specimens investigated in this study possess sub-triangular-shaped nasal sacs (nc, Figs. 5 and 7). The olfactory nerve canals (nI) exit the nasal sac postero-dorsally and are large (nI, Figs. 3–8; Table S2). Lateral to the point at which nI exits the nasal sac, another canal emerges postero-laterally, which is interpreted as the orbitonasal canal (c.on, Figs. 3–8; labelled V2, figs. 65 & 66 in Miles, 1977). This passes postero-laterally for a short distance before entering the cavum epiptericum. Another canal emanates from the posterior margin of the nasal sac that would have housed the medial branch of the nasal vein. This canal bifurcates into posteriorly and postero-laterally directed branches (l.v.na, m.v.na, Figs. 3–8). The more lateral of these is the canal for the lateral branch of the nasal vein, which continues postero-laterally yet also branches again back postero-medially rejoining the medial branch (l.v.na, Figs. 5–8). The medial branch continues into the common chamber for the palatine artery, ramus palatinus and nasal vein (co.ch, Figs. 7 and 8). From this chamber the canal for the ramus palatinus VII (which also houses the palatine artery) continues postero-medially, travelling towards the hypophyseal recess (r.pal.VII, Figs. 7 and 8) in accordance with the position in Miles (VIIp, fig. 64, Miles, 1977). Dorsal to the canal for the lateral branch of the nasal vein and the orbitonasal canal sits the antero-posteriorly directed median branch of the ramus opthalmicus profundus V (r.op.p.V, Figs. 6–8) which passes over the nasal sac. The configuration of the orbitonasal canal and ramus opthalmicus profundus V confirm prior interpretations (Miles, fig. 66, 1977), with interpretation of the other ethmoidal canals herein building upon that of Miles (1977).

Figure 3 Dorsal aspect of the endocast of ‘Chirodipterus’ australis specimen NHMUK PV P56035.

Digital 3D endocast of ‘Chirodipterus’ australis specimen NHMUK PV P56035. Scale bar = 10 mm. (A) Dorsal aspect of the digital endocast with arteries in red, veins in blue and cranial cavity and nerves in yellow. (B) Interpretive drawing with characters discussed in the text labelled. Abbreviations: amp.ascc, ampulla of the anterior semicircular canal; amp.lscc, ampulla of the lateral semicircular canal; ascc, anterior semicircular canal; b.olf, olfactory bulb; c.a.opth, canal for the opthalmic artery; c.on, orbitonasal canal; c.v.acb, canal for the anterior cerebral vein; c.v.jug, canal for the jugular vein; c.v.mcv, canal for the middle cerebral vein; c.v.pit, canal for the pituitary vein; cav.so, supraotic cavity; d.end, endolymphatic duct; lscc, lateral semicircular canal; m.v.na, canal for the median branch of the nasal vein; nI, canal for the olfactory nerve; nII, canal for the optric nerve II; nIII, canal for the occulomotor nerve III; nIV, canal for the trochlear nerve IV; nV1, canal for the opthalmicus profundus nerve V1; nV2, V3, canal for the maxillaris nerve V2 and the mandibularis nerve V3; nVII, canal for the facial nerve VII; nc, nasal capsule; pscc, posterior semicircular canal; rec.pp, recess for the pineal-parapineal recess; sp.occ.1–4, canals for spino-occipital nerves 1–4.

Figure 4 Right lateral aspect of the endocast specimen of ‘Chirodipterus’ australis specimen NHMUK PV P56035.

Digital 3D endocast of ‘Chirodipterus’ australis specimen NHMUK PV P56035. Scale bar = 10 mm. (A) Right lateral aspect of the digital endocast with arteries in red, veins in blue and cranial cavity and nerves in yellow. (B) Interpretive drawing with characters discussed in the text labelled. Abbreviations: amp.ascc, ampulla of the anterior semicircular canal; amp.lscc, ampulla of the lateral semicircular canal; amp.pscc, ampulla of the posterior semicircular canal; ascc, anterior semicircular canal; b.olf, olfactory bulb; c.a.opth, canal for the opthalmic artery; c.a.ic, canal for the internal carotid artery; c.bv, canal for a small blood vessel; c.on, orbitonasal canal; c.v.acb, canal for the anterior cerebral vein; c.v.jug, canal for the jugular vein; c.a.occ, canal for the occipital artery; c.v.pit, canal for the pituitary vein; d.end, endolymphatic duct; lscc, lateral semicircular canal; m.v.na, canal for the median branch of the nasal vein; nI, canal for the olfactory nerve; nII, canal for the optric nerve II; nIII, canal for the occulomotor nerve III; nIV, canal for the trochlear nerve IV; nV1, canal for the opthalmicus profundus nerve V1; nV2, V3, canal for the maxillaris nerve V2 and the mandibularis nerve V3; nVII, canal for the facial nerve VII; nIX, canal for the glossopharyngeal nerve IX; nX, canal for the vagus nerve X; nc, nasal capsule; nt, notochord; pscc, posterior semicircular canal; rec. hyp, recess for the hypophyseal recess; rec.pp, recess for the pineal-parapineal recess; rec.sacc.lag, recess for the sacculolagenar pouch; rec.utr, utricular recess; sc, spinal cord; sp.occ.4, canal for spino-occipital nerve 4.

Figure 5 Ventral aspect of the endocast of ‘Chirodipterus’ australis specimen NHMUK PV P56035.

Digital 3D endocast of ‘Chirodipterus’ australis specimen NHMUK PV P56035. Scale bar = 10 mm. (A) Ventral aspect of the digital endocast with arteries in red, veins in blue and cranial cavity and nerves in yellow. (B) Interpretive drawing with characters discussed in the text labelled. Abbreviations: amp.lscc, ampulla of the lateral semicircular canal; amp.pscc, ampulla of the posterior semicircular canal; b.olf, olfactory bulb; c.a.opth, canal for the opthalmic artery; c.a.ic.pb, canal for blood vessel which branches into the internal carotid artery and the pseudobranchial artery; c.on, orbitonasal canal; c.v.acb, canal for the anterior cerebral vein; c.v.mcv, canal for the middle cerebral vein; c.v.jug, canal for the jugular vein; c.a.occ, canal for the occipital artery; c.v.pit, canal for the pituitary vein; l.v.na, canal for the lateral branch of the nasal vein; lscc, lateral semicircular canal; m.v.na, canal for the median branch of the nasal vein; nI, canal for the olfactory nerve; nII, canal for the optric nerve II; nV1, canal for the opthalmicus profundus nerve V1; nV2, V3, canal for the maxillaris nerve V2 and the mandibularis nerve V3; nVII, canal for the facial nerve VII; nIX, canal for the glossopharyngeal nerve IX; nX, canal for the vagus nerve X; nc, nasal capsule; nt, notochord; rec. hyp, recess for the hypophyseal recess; rec.sacc.lag, recess for the sacculolagenar pouch; sp.occ.1, canal for spino-occipital nerve 1; ven.exp.t, ventral expansion of the telencephalon.

Figure 6 Dorsal aspect of the endocast of ‘Chirodipterus’ australis specimen NHMUK PV P56038.

Digital 3D endocast of ‘Chirodipterus’ australis specimen NHMUK PV P56038. Scale bar = 10 mm. (A) Dorsal aspect of the digital endocast with arteries in red, veins in blue and cranial cavity and nerves in yellow. (B) Interpretive drawing with characters discussed in the text labelled. Abbreviations: amp.lscc, ampulla of the lateral semicircular canal; amp.pscc, ampulla of the posterior semicircular canal; ascc, anterior semicircular canal; b.olf, olfactory bulb; bif.nI, bifurcation in the canal for the olfactory nerve I; c.a.ic.pb, canal for blood vessel with branches into the internal carotid artery and the pseudobranchial artery; c.a.orb, canal for the orbital artery; c.on, orbitonasal canal; c.v.acb?, canal possibly for the anterior cerebral vein; c.v.jug, canal for the jugular vein; c.v.mcv, canal for the middle cerebral vein; cav.so, supraotic cavity; d.end, endolymphatic duct; l.v.na, canal for the lateral branch of the nasal vein; lscc, lateral semicircular canal; m.v.na, canal for the median branch of the nasal vein; nII, canal for the optric nerve II; nIV, canal for the trochlear nerve IV; nV1, canal for the opthalmicus profundus nerve V1; nV2, V3, canal for the maxillaris nerve V2 and the mandibularis nerve V3; nVI, canal for the abducens nerve VI; nVII, canal for the facial nerve VII; nX, canal for the vagus nerve X; nc, nasal capsule; pscc, posterior semicircular canal; r.op.p.V, canal for ramus opthalmicus profundus nV; rec.pp, recess for the pineal-parapineal recess; sp.occ.2d, canal for the dorsal division of spino-occipital nerve 2; sp.occ.3–4, canals for spino-occipital nerves 3–4.

Figure 7 Right lateral aspect of the endocast of ‘Chirodipterus’ australis specimen NHMUK PV P56038.

Digital 3D endocast of ‘Chirodipterus’ australis specimen NHMUK PV P56038. Scale bar = 10 mm. (A) Right lateral aspect of the digital endocast with arteries in red, veins in blue and cranial cavity and nerves in yellow. (B) Interpretive drawing with characters discussed in the text labelled. Abbreviations: amp.ascc, ampulla of the anterior semicircular canal; amp.lscc, ampulla of the lateral semicircular canal; amp.pscc, ampulla of the posterior semicircular canal; ascc, anterior semicircular canal; b.olf, olfactory bulb; c.a.ic, canal for the internal carotid artery; c.a.opth, canal for the opthalmic artery; c.on, orbitonasal canal; c.a.pb, canal for the pseudobranchial artery; c.v.acb?, canal possibly for the anterior cerebral vein; c.v.jug, canal for the jugular vein; co.ch, common chamber to palatine artery, orbitonasal canal, canal for the ramus palatinus nVII and the canal for the medial nasal vein; d.end, endolymphatic duct; lscc, lateral semicircular canal; m.v.na, canal for the median branch of the nasal vein; nII, canal for the optric nerve II; nIV, canal for the trochlear nerve IV; nV1, canal for the opthalmicus profundus nerve V1; nV2, V3, canal for the maxillaris nerve V2 and the mandibularis nerve V3; nVII, canal for the facial nerve VII; nIX, canal for the glossopharyngeal nerve IX; nX, canal for the vagus nerve X; nc, nasal capsule; pscc, posterior semicircular canal; r.op.p.V, canal for ramus opthalmicus profundus nV; r.pal.VII, canal for the ramus palatinus nVII; rec. hyp, recess for the hypophyseal recess; rec.pp, recess for the pineal-parapineal recess; rec.sacc.lag, recess for the sacculolagenar pouch; rec.utr, utricular recess; sp.occ.2d.2v, canal for spino-occipital nerve 2, splitting into dorsal and ventral divisions; sp.occ.4, canal for spino-occipital nerve 4; ven.exp.t, ventral expansion of the telencephalon.

Figure 8 Ventral aspect of the endocast of ‘Chirodipterus’ australis specimen NHMUK PV P56038.

Digital 3D endocast of ‘Chirodipterus’ australis specimen NHMUK PV P56038. Scale bar = 10 mm. (A) Ventral aspect of the digital endocast with arteries in red, veins in blue and cranial cavity and nerves in yellow. (B) Interpretive drawing with characters discussed in the text labelled. Abbreviations: amp.lscc, ampulla of the lateral semicircular canal; bif.nI, bifurcation in the canal for the olfactory nerve I; c.a.ic, canal for the internal carotid artery; c.a.pb canal for the pseudobranchial artery; c.a.opth, canal for the opthalmic artery; c.a.orb, canal for the orbital artery; c.on, orbitonasal canal; c.v.mcv, canal for the middle cerebral vein; c.v.jug, canal for the jugular vein; co.ch, common chamber to palatine artery, orbitonasal canal, canal for the ramus palatinus nVII and the canal for the medial nasal vein; l.v.na, canal for the lateral branch of the nasal vein; lscc, lateral semicircular canal; m.v.na, canal for the median branch of the nasal vein; nII, canal for the optric nerve II; nV1, canal for the opthalmicus profundus nerve V1; nV2, V3, canal for the maxillaris nerve V2 and the mandibularis nerve V3; nVI, canal for the abducens nerve VI; nVII, canal for the facial nerve VII; nIX, canal for the glossopharyngeal nerve IX; nX, canal for the vagus nerve X; nc, nasal capsule; r.op.p.V, canal for ramus opthalmicus profundus nV; r.pal.VII, canal for the ramus palatinus nVII; rec. hyp, recess for the hypophyseal recess; rec.sacc.lag, recess for the sacculolagenar pouch; sp.occ.1, canal for spino-occipital nerve 1; sp.occ.2d.2v, canal for spino-occipital nerve 2, splitting into dorsal and ventral divisions; sp.occ.3d.3v, canal for spino-occipital nerve 3, splitting into dorsal and ventral divisions; sp.occ.4, canal for spino-occipital nerve 4; ven.exp.t, ventral expansion of the telencephalon.

Olfactory nerve canals: Rising posteriorly from the nasal sacs, the olfactory nerve canals extend to where they meet the anterior margin of the telenecephalon (nI, Figs. 3–5; Table S2). There is a slight swelling of nI prior to the union with the forebrain (Table S2) which are here interpreted as sessile olfactory bulbs (b.olf, Figs. 3–7). Olfactory tracts, or peduncles, are absent in ‘Chirodipterus’ australis. The olfactory nerve canals project ventrally at a greater angle in NHMUK PV P56035 than they do in NHMUK PV P56038 (Table S2), most likely due to the dorso-ventral compression of NHMUK PV P56038. The olfactory nerves also diverge from each other at the rostral margin of the forebrain (Table S2).

From the juncture of the olfactory bulbs and the telencephalon the anterior cerebral vein emanates in a dorso-lateral direction before descending ventro-laterally in NHMUK PV P56035 (c.v.acb, Figs. 3–5). A single canal emanates from the point at which the olfactory tracts diverge in NHMUK PV P56038, and this may be a branch of the anterior cerebral vein (c.v.acb?, Figs. 6 and 7); however, tiny bumps can be seen on the dorsal surface of the olfactory nerve canals in this specimen in a comparable position to those in NHMUK PV P56035, though it is unclear if they represent the true root of the anterior cerebral veins (c.v.acb?, Fig. 6). If the single medial canal anterior to the pineal in NHMUK PV P56038 is not the anterior cerebral canal, it is an structure unobserved in any other dipnoan endocast. A small canal emerges from the ventral margin of the telencephalon, anterior to the ventral expansion of the telencephalon, which has not been observed in any dipnoan cranial endocast before and may have been a canal for a small blood vessel (c.bv, Fig. 4). No bifurcation of the olfactory nerve canals is seen at, or prior to, the meeting with the nasal sac in NHMUK PV P56035 as seen in Dipterus. Bifurcation is, however, observed in both the left and right olfactory tracts of NHMUK PV P56038 anterior to the olfactory bulbs (bif.nI, Figs. 6 and 8).

Forebrain (telencephalon & diencephalon): Directly posterior to the junction of the olfactory nerve canals and the forebrain there is a ventral expansion of the telencephalon (Table S1), which steeply recedes at its posterior end (ven.exp.t, Figs. 5 and 7). Just dorsal to the posterior margin of the ventral expansion there is a large canal of nerve II (nII) approximately 3 mm posterior to the olfactory bulbs, which extends antero-laterally (nII, Figs. 3–8). Dorsal to nII in NHMUK PV P56035, and postero-dorsal to nII in NHMUK PV P56038, lies another canal emanating dorso-laterally from the forebrain. This canal would have housed the oculomotor nerve III (nIII, Figs. 3, 4, 6 and 7).

On the dorsal surface of the diencephalon, the shared pineal-parapineal recess is directed antero-dorsally and does not penetrate the dermal bones (rec.pp, Figs. 3, 4, 6 and 7). Posterior to the pineal-parapineal recess, the dorsal surface is flat for approx. 5 mm before rising gradually. On the ventral side of the diencephalon, posterior to nII, the hypophyseal recess extends ventrally (rec.hyp, Figs. 4, 5 and 7–9). Some of the hypophyseal region could not be segmented due to damage in the ventral portion of the cranial cavity, however a significant portion can be resolved. The segmented region reveals paired thin canals emerging antero-laterally from the anterior lateral margin of the recess, which housed the opthalmic arteries (c.a.opth, Figs. 3–5, 7–9). Posterior to the origin of these canals, paired canals leave the hypophyseal recess postero-laterally. These posterior canals continue for 4.5 mm before turning dorsally 22°and bifurcating into the ventro-medially directed internal carotid artery and the postero-dorsally directed pseudobranchial artery (c.a.ic.pb, Figs. 5, 6, 8 and 9; c.a.ic, Figs. 4 and 7; c.a.pb, Fig. 8). The pseudobranchial artery joins another canal, which originates ventral to the facial nerve (nVII) in the mesencephalon (described below). Another pair of canals are observed directed anteriorly from a more medial position than the opthalmic arteries. These do not emanate from the hypophyseal recess however, but intersect the common chamber for the nasal veins and are interpreted as canals for the ramus palatinus VII and the palatine artery (r.pal.VII, Figs. 7 and 8; see also fig. 4, Challands, 2015).

Figure 9 The hypophyseal recess of NHMUK PV P56035.

(A) Right lateral and (B) left lateral view of the hypophyseal recess of ‘Chirodipterus’ australis. Abbreviations as in Figs. 3–8.

Dorsal to the hypophyseal recess of NHMUK PV P56035 is the canal for the pituitary vein, which extends antero-ventro-laterally (c.v.pit, Figs. 3–5 and 9). On the right of the endocast, anterior and dorsal to the canal for the pituitary vein, another canal is directed laterally and anteriorly, housing nerve IV (nIV, Figs. 3 and 4).

Midbrain (mesencephalon): The transition from the diencephalon to the mesencephalon in ‘C.’ australis is indicated by a steady posterior incline on the dorsal surface of the endocast, which is accompanied by lateral expansion of the endocast. On the ventral margin of the endocast, the mesencephalon can be seen to begin at the posterior limit of the hypophyseal recess. The beginning of the mesencephalon has been associated with a rise in the brain roof from the optic tectum, which is also signified by lateral expansion of the endocast (Northcutt, 1986; Nieuwenhuys, 2014). Furthermore, the endocast possesses a large dorsal swelling at the pinnacle of the dorsal mesencephalon, indicating the presence of optic lobes (optic lobe, Figs. 4 and 7).

The canal for the trigeminal nerve complex emanates ventral and lateral to the pinnacle of the mesencephalon, posterior to the canal for the pituitary vein. On both sides of the endocast, this canal bifurcates into one canal travelling anteriorly and laterally, for the ophthalmicus profundus nerve (nV1), and another postero-laterally, which held both the maxillaris nerve (nV2) and the mandubularis nerve (nV3; nV1 and nV2, nV3, Figs. 3–8). The canal for the ophthalmicus profundus (nV1) continues antero-laterally until joining the canal for the jugular vein. Postero-dorsal to the origin of the trigeminal complex is another canal projecting postero-laterally, that would have housed the facial nerve (nVII, Figs. 3–8). This canal remains separated from the posterior branch of the trigeminal initially, yet the two soon merge to form one large laterally directed canal (Fig. 3). Union of these canals suggests that nVII, nV2 and nV3 were housed in the same canal as they left the neurocranium, though loss of a dividing wall between these canals during fossilisation could produce such a structure as an artefact. The former interpretation is supported by observations made by Miles (fig. 17, 1977) who recognised that these canals are separated, yet exit the neurocranium via one large foramen (along with the canal for nV1 before it joins the jugular canal). Posterior to the bifurcating canal of the trigeminal complex is another canal directed postero-laterally from the endocast (this canal is depicted in fig. 9 of Säve-Söderbergh, 1952, but is not labelled, and is labelled by Stensiö, 1963 as a “canal probably for a ... middle cerebral vein?”). On the right side of the endocast of NHMUK PV P56038, this canal continues for 3.5 mm before meeting the canal for the pseudobranchial artery, which originates at the hypophyseal recess, and ultimately the canal for the jugular vein (c.v.mcv, Figs. 3, 5, 6 and 8; nVI, Figs. 6 and 8). This canal likely represents that for the middle cerebral vein as suggested by Stensiö (1963) and Miles (fig. 47, 1977) rather than the canal for the abducens nerve nVI as interpreted by Challands (2015) for a similar structure in Dipterus. Furthermore, in NHMUK PV P56038, a separate canal also emerges from the main body of the endocast posterior and ventral to nerve VII and the canal for the middle cerebral vein. This likely represents the true root of the abducens (nVI) as seen in the endocast of a fossil lungfish and the identification made by Challands (2015) appears to have been made in error.

A branch of the orbital artery runs alongside and slightly dorso-lateral to the jugular vein (c.a.orb, Figs. 6 and 8). Anteriorly, however (in line with the anterior margin of the labyrinth), the orbital artery passes over the dorsal surface of the jugular vein. The observed layout here confirms the configuration noted in Miles (1977), where the jugular vein, orbital artery and the nerves of the trigeminal complex are represented by separate foramina in the cavum epiptericum.

Hindbrain (rhombencephalon): The supraotic cavities are small, however large endolymphatic ducts emerge from them postero-dorsally (cav.so, Figs. 3 and 6; d.end, Figs. 3, 4, 6 and 7; Table S2). The canal for the glossopharyngeal nerve (nIX) emanates from the posterior region of the sacculus and is directed postero-ventrally, ventral to the junction of the lateral and posterior semi-circular canals and dorsal to the posterior limit of the sacculolagenar (nIX, Figs. 4, 5, 7 and 8). It does not divide as suggested by Miles (1977).

Dorso-medially to nIX, a large canal directed postero-ventrally exits the hindbrain which, then joins another large canal that originates in the otic region posterior and ventral to the endolymphatic ducts and proceeds postero-ventro-laterally. The union of these two canals lies postero-medially to the meeting of the posterior and lateral semicircular canals, although it is separated from the ampullae. These two canals are interpreted as confirmation of the separate dorsal and ventral divisions of the vagus nerve (nX, Figs. 4–8) outlined by Miles (p. 38–43, fig. 15, 1977). The divisions are separated by what Miles termed the paravagal process, with vascular tissue and blood vessels, notably the posterior cerebral vein, housed in the dorsal division and the ventral division containing nervous tissue. Level with and posterior to the origin of the nervous division of nX, the first spino-occipital nerve exits the spinal cord postero-laterally (sp.occ.1, Figs. 3, 5 and 8). Dorsal and posterior to the first spino-occipital nerve is the second spino-occipital nerve, which bifurcates into dorsally and ventrally directed branches exiting the neurocranium (sp.occ.2d.2v, Figs. 7 and 8), as figured by Miles (sp.occ.2d and sp.occ.2v, fig. 15, 1977). Further dorsally lies the third spino-occipital nerve, which follows a similar configuration to the second, with dorsal and ventral divisions (sp.occ.3d.3v, Figs. 3, 6 and 8; sp.occ3d and sp.occ3v, fig. 15, Miles, 1977).

Ventral to the spinal cord (sc, Fig. 4) is the notochord (nt, Figs. 4 and 5). These two structures are separated from each other by an ossified shelf shown by the separated spinal cord and notochord in the posterior region of the hindbrain. The notochord appears to merge with the sacculi in the endocast, though the two would have been separate in life, and extends anteriorly as far forward as the anterior limit of the sacculolagenar, providing an exact description of the anterior margin of the notochord which Miles (1977) was unable to discern. The notochord would, however, have had a thick lining of cartilage in life, meaning that the canal observed herein will be slightly enlarged compared to the true size. The occipital artery originates from the posterior union of the hindbrain and labyrinth running postero-ventro-laterally before turning medially to pass closely to the lateral margins of the notochord (c.a.occ, Figs. 4 and 5).

Labyrinth: The labyrinth system of ‘Chirodipterus’ australis has previously been figured by Miles (figs. 47 & 48, 1977). The endocast described here is in agreement with Miles regarding the relative dimensions and positioning of the semicircular canals, utricular recess and sinus superior, however the sacculi are presented here in greater detail.

The crus commune, where the anterior and posterior semicircular canals meet, is lower than the dorsal-most point of both the anterior and posterior semicircular canals (ascc, Figs. 3, 4, 6 and 7; pscc, Figs. 3, 4, 6 and 7; Table S4). The sinus superior does, however, extend dorsally beyond the roof of the hindbrain. There is a slight dorsal expansion where the sinus superior meets the hindbrain, anterior to the base of the endolymphatic ducts, which designates an anterior division of the supraotic cavity, despite previous indications that ‘Chirodipterus’ australis lacks such a division (Miles, 1977).

There is no observable deflection of the anterior semi-circular canal (SCC), yet there is a large ampulla at its antero-ventral margin before it communicates with the lateral SCC (amp.ascc, Figs. 3, 4 and 7; Table S4). Just posterior and lateral to this junction, the lateral SSC also exhibits a large ampulla (amp.lscc, Figs. 3–8; Table S4). These two canals intersect with the dorsal surface of the utricular recess, which is small (Table S4) and extends laterally from the sacculolagenar (rec.utr, Figs. 4 and 7). The posterior SCC also exhibits an ampulla at the intersection with the lateral SCC (amp.pscc, Figs. 4–7; Table S4), a feature not clearly depicted by Miles (1977). This character was coded in Friedman (2007) stating that ‘Chirodipterus’ australis lacks a major expansion at the intersection of the posterior SCC and the sacculus. Here, however, there appears to be expansion in this region (Table S4). Also contra Miles (1977), the sacculus of ‘C.’ australis is slightly elongated and ovoid, extending both anteriorly and posteriorly (rec.sacc.lag, Figs. 4, 5, 7 and 8). The anterior margin of the sacculus extends beyond that of the utriculus, unlike the depiction in Miles (fig. 48, 1977), and posteriorly, the sacculi extend until they are vertically in line with the posterior margin of the lateral SCC. The bone on the base of the sacculus is very thin, providing poor density contrast. Problems with density contrast in segmentation of the endosseous labyrinth have been highlighted by Walsh, Luo & Barrett (2013); however, the margins of the sacculus were identifiable and show a relatively simple rounded shape similar to modern lungfish (Jorgensen & Joss, 2010). No notch is observed separating the sacculus and lagena as in Rhinodipterus kimberleyensis and Dipterus.

Discussion

Morphological comparisons with other taxa

Endocasts of fossil Devonian dipnoans available for comparison with ‘Chirodipterus’ australis include Chirodipterus wildungensis (Säve-Söderbergh, 1952), Dipnorhynchus sussmilchi (Campbell & Barwick, 1982; Clement et al., 2016), Rhinodipterus kimberleyensis (Clement & Ahlberg, 2014) and Dipterus valenciennesi (Challands, 2015). Comparison of the endosseous labyrinth is possible with Griphognathus whitei (Miles, 1977, fig. 46), as is comparison with the endocast of the probable stem-dipnoan Youngolepis praecursor (Chang, 1982). Furthermore, comparison of the endocasts herein with the previous endocast interpretation of ‘Chirodipterus’ australis by Miles (1977) is also possible.

‘Chirodipterus’ australis displays lateral compression of the fore- and midbrain, and small optic lobes characteristic of fossil dipnoan endocasts (Stensiö, 1963; Northcutt, 1986). Similar to Dipterus valenciennesi, the ventral margins of the diencephalon, mesencephalon and metencephalon are not deep, though the hypophyseal recess is well developed, similar to the osteolepid condition exemplified by Eusthenopteron foordi (Stensiö, 1963, fig. 50A). The endocast of Chirodipterus wildungensis (Säve-Söderbergh, 1952) appears to be further expanded ventrally in these divisions of the endocast. It is possible that the reconstruction in C. wildungensis was influenced by the brain of the extant Neoceratodus, and perhaps also impacted by poor preservation of the fossil material. This latter factor is important in comparison with the digital endocast of Rhinodipterus kimberleyensis (Clement & Ahlberg, 2014), in which dorsal and ventral expansion of the midbrain may be an artefact of preservation; the fidelity of the data in this region is poor.

Nasal sac: Unlike Dipnorhynchus sussmilchi, in which the nasal sac is directed medially and is elongated antero-posteriorly (Thomson & Campbell, 1971, fig. 29), the sub-triangular nasal sac of ‘C.’ australis extends laterally as well as anteriorly similar to Dipterus valenciennesi (Challands, 2015), Porolepis spitsbergensis (Jarvik, 1972) and Youngolepis praecursor (Chang, 1982). The nasal sac of ‘C.’ australis also differs from that of G. whitei, with its defined solum nasi and posterior opening, as described by Miles (1977). Comparison with the ventral depiction of the anterior cranial cavity of C. wildungensis (Säve-Söderbergh, 1952, fig. 3) shows that this species may have had nasal sacs resembling ‘C.’ australis in that they are large and open ventrally, though their exact shape is not described.

Within Devonian Dipnoi, the positioning of the orbitonasal canal is identical to that in Holodipterus gogoensis (Miles, 1977, figs. 69 & 70) and similar to G. whitei, in which it enters the posterior nasal sac (labelled V2, Miles, 1977, fig. 63). Furthermore, the condition in C. wildungensis appears similar to ‘C.’ australis in that it enters the nasal sac, though it enters ventral to the opening in the nasal sac for the olfactory nerve (Säve-Söderbergh, 1952), not laterally. The orbitonasal canal in ‘C.’ australis is also similar in positioning to that of Porolepis spitsbergensis, Youngolepis praecursor and Powichthys spitsbergensis, three Lower Devonian dipnomorphs, in which the orbitonasal enters the nasal sac posteriorly (Jarvik, 1972, fig. 14; Chang, 1982, fig. 14d; Clément & Ahlberg, 2010, fig. 7f), though this canal is much smaller in these taxa than ‘C.’ australis. Ahlberg (1991) suggested that the difference in size may be due to the vein transmitted in the orbitonasal, along with the ramus maxillaris nV2, in dipnoans was actually contained within the large profundus nerve canal in porolepiforms and primitive dipnomorphs. ‘C.’ australis is, however, unlike Dipterus valenciennesi, in which the orbitonasal canal circumvents the nasal sac antero-medially albeit with a small branch perforating the nasal sac posteriorly (Challands, 2015, fig. 4), and Dipnorhynchus sussmilchi, in which the orbitonasal canal joins the palatine artery (labelled the groove for the subnasal vein, gr. subn. v, and f. pal. VII, Thomson & Campbell, 1971, fig. 29; this structure was not segmented in Clement et al., 2016). Guiyu oneiros, a primitive sarcopterygian, displays medial positioning of the orbitonasal canal (Qiao & Zhu, 2010). It therefore appears that primitive dipnomorphs and Upper Devonian dipnoans possess orbitonasal canals that enter the nasal sac, whereas in Dipterus and Dipnorhynchus (Lower and Middle Devonian dipnoans) the nasal sac is circumvented, with the caveat that the reconstruction of Thomson & Campbell (1971) is correct. Thus, a puzzling distribution of this character is presented, and it seems that the topological morphology of the orbitonasal canal is largely conservative within the Dipnomorpha, with deviation in the two early dipnoans (Dipnorhynchus and Dipterus), though there is potential that the soft tissue structures carried by the canal change in the Dipnoi. Dipterus valenciennesi is therefore found to be unique with regard to the orbitonasal, in that it has ossified in a separate canal outwith the nasal sac, rather than following an antero-medial path inside the nasal sac as in ‘Chirodipterus’ australis and Holodipterus gogoensis (Miles, 1977, figs. 65 & 70). Unfortunately, no endocast of primitive sarcopterygians outside the Dipnomorpha have exhibited an orbitonasal canal, though with future studies this may prove to be an informative character.

The canal for the ramus opthalmicus profundus V in ‘C.’ australis resembles all other Upper Devonian dipnoans (Säve-Söderbergh, 1952, fig. 4; Miles, 1977, figs. 63 & 69), Dipterus valenciennesi (Challands, 2015, fig. 4), and also Youngolepis praecursor (Chang, 1982, figs. 14 & 17), in passing dorsally over the nasal sac. This is unlike the condition observed in Dipnorhynchus kurikae (Campbell & Barwick, 2000, fig. 7) and Dipnorhynchus sussmilchi (Thomson & Campbell, 1971, fig. 29; Clement et al., 2016, fig. 3), where this canal joins the nasal sac posteriorly; this same state is observed in the Porolepiformes and Powichthys (Jarvik, 1972; Clement and Ahlberg, 2010). That this union of the ramus opthalmicus profundus V and the nasal sac is contained within the monophyletic Dipnorhynchus (within Dipnoi), while all other dipnoans share the state of Youngolepis praecursor, suggests that separation of the two structures is the ancestral state of stem-group dipnoans and that it is an autapomorphy in Dipnorhynchus (Challands, 2015). The primitive dipnomorph state is at this time unclear due to the divide between Porolepiformes + Powichthys and Dipnoi + Youngolepis, the two hypothesised branches of the Dipnomorpha. Determination of the configuration of this character in other basal dipnoans, such as Diabolepis and Uranolophus, as well as basal sarcopterygians may resolve the polarity of this character and allow for it to be coded confidently in phylogenetic analyses.

The nasal vein in ‘C.’ australis appears to be in close agreement with that of Dipterus valenciennesi (Challands, 2015, fig. 4). Both possess a common chamber for the palatine artery, medial nasal vein and ramus palatinus VII; however, they differ in that ‘C.’ australis has only one emanating vein from the posterior nasal sac where Dipterus valenciennesi has three. This is quite possibly due to the fact that, in ‘C.’ australis, there are two unions of the lateral branch of the nasal vein and the medial branch prior to the meeting with the nasal sac—lateral and anterior to the common chamber—whereas in Dipterus valenciennesi there is just one, lateral to the common chamber. Such differences may be explained by the fact that minor changes in ossification in this region could easily shift the position of canals and, particularly where canals are in such close proximity, cause the union of multiple canals. It is also possible for these venous canals to be absorbed into the orbitonasal canal in species where the orbitonasal joins the nasal sac, or in Dipnorhynchus, into either the canal for the ramus opthalmicus profundus V or the ramus opthalmicus mandibularis V. Determination of the configuration of such canals within other dipnoans may find that the number and identity of canals entering the nasal sac posteriorly could be used as an additional character in phylogenetic study though this requires detailed segmentation of high quality μCT data.

Olfactory nerve canals: The bifurcation of the olfactory nerve canals prior to the nasal sac now observed in ‘C.’ australis is a characteristic shared, among the Dipnoi, only with Dipterus valenciennesi (Challands, 2015, figs. 6 & 7) and Dipnorhynchus (sussmilchi—described in text, Thomson & Campbell, 1971; kurikae—Campbell & Barwick, 2000, fig. 6b). With the most recent phylogenies placing ‘C.’ australis in ever more basal positions, it is worth taking into consideration that no endocast of any dipnoan between Dipnorhynchus and ‘C.’ australis other than Dipterus valenciennesi has been described (Clement et al., 2016). Miles (1977) did not identify a bifurcating canal in Gogodipterus paddyensis, often placed as the sister taxon to ‘C.’ australis, although this was also the case in his more detailed examination of ‘C.’ australis, which included the specimens used here. This proves that characters can remain elusive even under rigorous scrutiny of fossil material, particularly those concerned with a thin separating layer of bone. R. kimberleyensis has been μCT scanned, however the specimen was missing the snout tip and the anterior reaches of the olfactory nerve canals could not be examined (Clement & Ahlberg, 2014). Bifurcation was not observed in the endocast of C. wildungensis (Säve-Söderbergh, 1952), though this feature would unlikely manifest in a physical endocast, and definitely not where the specimen is incomplete (Miles, 1977). From this analysis, we conclude that bifurcating olfactory nerve canals are a primitive character in Dipnoi, and that extant forms exhibit the derived state. The basal dipnomorph Youngolepis praecursor displays a small bifurcation in the right olfactory nerve canal (Chang, 1982, fig. 18); however, neither the porolepiform Porolepis spitsbergensis nor the osteolepiform Eusthenopteron foordi (Jarvik, 1972, fig. 69) do. These observations may again be inaccurate due to imperfect preservation in the rostral region of the specimens examined or actually indicate that this character is autapomorphic to the Dipnoi and Youngolepis, lending further weight to the hypothesis that Youngolepis is a stem-dipnoan. The digital endocast of Tungsenia paradoxa, an Early Devonian tetrapodomorph, also showed no such bifurcation in its short olfactory nerve canals which otherwise resemble basal dipnomorphs (Lu et al., 2012), further suggesting that bifurcation may be autapomorphic to dipnoans inclusive of Youngolepis. It is clear that except in circumstances where exceptional preservation is combined with μCT scanning this character is difficult to identify, and as such validation of this as an informative character will require further research in taxa lying basal to ‘C.’ australis.

The orientation of the olfactory nerve canals in NHMUK PV P56035 deserves brief discussion. Downward projecting olfactory canals at such a high angle as seen in ‘C.’ australis has not been observed in the endocast of any dipnoan. The dorso-ventral compression of NHMUK PV P56038 prevented accurate determination of this trait, and neither the account of C. wildungensis (Säve-Söderbergh, 1952) nor (Miles, 1977) description of ‘C.’ australis mention descending olfactory nerve canals. The perfect three-dimensional nature of specimen NHMUK PV P56035, however, precludes this feature being an artefact of compression or preservation. While at this stage of our understanding in the relationship between neurocrania and endocast accounting for such a structure is speculative, it is possible that the ventrally directed olfactory tracts are a product of accommodating a short- and deep-snouted skull. Rather than secondarily shortening, they have become ventrally oriented to accommodate the same length in a shorter space. In contrast the olfactory canals of G. whitei are extremely elongated to account for the lengthening of the snout (Miles, 1977, fig. 63).

Telencephalon: ‘Chirodipterus’ australis possesses some ventral expansion of the telencephalon, unlike the basal Youngolepis praecursor and Powichthys spitsbergensis, which lack expansion of this region (Chang, 1982; Clement and Ahlberg, 2010). Extant lungfish, Neoceratodus and the Lepidosireniformes, both exhibit extreme expansion of the telencephalon (Northcutt, 1986), as does Rhinodipterus kimberleyensis, though not to the extent of extant taxa (Clement & Ahlberg, 2014), while Dipterus valenciennesi (Challands, 2015) and Dipnorchynchus sussmilchi (Clement et al., 2016) possess very slight ventral expansion These observations have confirmed a trend of gradually increasing ventral expansion in Dipnoi. It has been inferred that this is due to an increasing volume of the subpallium, related to olfaction, corresponding with the evolution of the palatal bite and autostyly (Clement & Ahlberg, 2014), however it may also mirror the change seen in chondrichthyans which corresponds to environmental or social navigational capacity (Yopak et al., 2007; Clement et al., 2016). The presence of ventral expansion, although only slight, in the most basal lungfish for which endocasts have been produced, though not in the dipnomorph Powichthys, proposes the hypothesis that this ventral expansion has its roots with the Dipnoi. Digital examination of Youngolepis praecursor and Diabolepis may prove interesting with regard to the origin of this character.

‘Chirodipterus’ australis has sessile olfactory bulbs, a trait shared with the Devonian dipnoans Dipterus valenciennesi (Challands, 2015), Rhinodipterus kimberleyensis (Clement & Ahlberg, 2014) and Dipnorhynchus sussmilchi (Clement et al., 2016), and with the extant Lepidosireniformes (Northcutt, 1986). The condition in C. wildungensis is unclear, with Säve-Söderbergh (1952) suggesting that the bulbs were lodged in the olfactory canals and therefore pedunculate, though Miles (1977) notes that it is equally plausible that they sat on the rostral portion of the telencephalon (sessile). The polarity of this character is still not unequivocal, because as yet, no Devonian dipnoan endocast has clearly displayed pedunculate olfactory bulbs. However, with Dipnorhynchus also displaying sessile olfactory bulbs the traditional view of pedunculate bulbs being the primitive state, which is based on the fact that they are observed in the coelacanth Latimeria chalumnae and the primitive extant lungfish Neoceratodus forsteri (Northcutt, 1986), must be questioned, particularly as they were not observed in the digital endocast of Neoceratodus (Clement et al., 2015). The current lines of evidence suggest that sessile bulbs are plesiomorphic to the Dipnoi. In the description of Youngolepis praecursor, Chang (1982) did not identify the olfactory bulbs, and so identification of this structure in in Youngolepis will further define polarity of this character in the Dipnomorpha. The porolepiforms Glyptolepis groenlandica and Porolepis spp. were, however, shown to have sessile olfactory bulbs (Jarvik, 1972, fig. 17), which suggests that sessile bulbs are plesiomorphic for the Dipnomorpha and that the pedunculate bulbs of Latimeria are apomorphic for Actinistia.

Also of note concerning the olfactory tracts is the relative positioning of the anterior cerebral vein. The position and form of this structure is variable within specimens (compare right and left side of the endocast in NHMUK PV P56035, Fig. 3), within species (between NHMUK PV P56035 and NHMUK PV P56038, Figs. 3 and 6) and between species (Dipterus valenciennesi, Challands, 2015, fig. 8). However, in no Devonian dipnoan does the anterior cerebral vein emanate from the lateral margin of the main body of the forebrain as depicted in C. wildungensis (Säve-Söderbergh, 1952, fig. 9). This would suggest that the endocast of C. wildungensis is expanded outwards beyond the true margin of the cranial cavity, at least in the region of the forebrain, and is therefore not truly representative of the shape of the endocast.

Diencephalon: The position of the optic (nII) and trochlear (nIV) nerves in ‘C.’ australis agree with that of C. wildungensis, although the canal for the oculomotor (nIII) is further posterior and ventral in C. wildungensis. Nerve III also exits the neurocranium of G. whitei, Dipnorhynchus sussmilchi and Dipterus valenciesnni similarly to C. wildungensis.

The shared pineal-parapineal recess of ‘C.’ australis is the derived condition, in contrast to the separate pineal and parapineal of Dipnorhynchus (Campbell & Barwick, 1982). Clement et al. (2016) did not recognise separate pineal and parapineal canals in their rendering of the endocast of Dipnorhynchus sussmilchi, but did recognise further medial canals posterior to the pineal canal. ‘C.’ australis lacks these dorso-medial canals posterior to the pineal canal. One additional medial canal is found in Dipterus valenciennesi, with Challands (2015) noting that Dipnorhynchus sussmilchi must be primitive in this regard, and Dipterus valenciennesi intermediate. The condition of this multistate character in R. kimberleyensis was unobservable due to damage in this region.

Despite there being damage to the hypophyseal recess of specimens examined in this study, it is clear the recess did not extend as far ventrally as that of Dipnorhynchus (Thomson & Campbell, 1971; Campbell & Barwick, 1982; Campbell & Barwick, 2000; Clement et al., 2016). Neither ‘C.’ australis, nor the two published endocasts of Devonian dipnoans Dipterus valenciennesi (Challands, 2015) and Rhinodipterus kimberleyensis (Clement & Ahlberg, 2014) exhibit such extension. Stensiö (1963) presentation of the brain and cranial nerves of fossil vertebrates shows that there was ventral projection of the hypophysis in Porolepiformes (Glyptolepis groenlandica, Jarvik, 1972, fig. 72a), Osteolepiformes (Eusthenopteron foordi, Jarvik, 1972, fig. 72b) and Actinistia (Latimeria), and the recent endocast of the onychodont Qingmenodus also displays ventral extension of the hypophysis (Lu et al., 2016). A common feature of all of these disparate sarcopterygian taxa is the presence of the hypophyseal foramen (also present in Youngolepis; Chang, 1982, fig. 7a), with the hypophyseal recess extending to pierce the parasphenoid. This foramen has been lost in derived Dipnoi, and as such the necessity for a ventrally-extended hypophysis is negated resulting in the shallow hypophyseal recesses of lungfish. Dipnorhynchus sussmilchi appears to exhibit an intermediate form, whereby the hypophyseal recess extends ventrally, though does not pierce the parasphenoid (Clement et al., 2016).

Unlike Dipterus valenciennsi (Challands, 2015) and Rhinodipterus kimberleyensis (Clement & Ahlberg, 2014), the pituitary vein of ‘C.’ australis emanates dorsal to the beginning of the hypophyseal recess. The exit of this canal from the neurocranium of C. wildungensis is in agreement with that of ‘C.’ australis. In comparing the endocast of the brain cavity with the true brain of Neoceratodus, the region surrounding the hypophysis was found to fit less closely than the forebrain and endosseous labyrinth (Clement et al., 2015). This may imply that the pituitary vein of ‘C.’ australis did emanate from the hypophyseal recess, as would be expected, but has not manifested in the endocast due to discrepancies between the morphology of the cranial cavity and the true extent of the brain.

The ophthalmic arteries are arranged as in Griphognathus whitei (Miles, 1977, figs. 35 & 56), Dipnorhynchus sussmilchi (Thomson & Campbell, 1971, fig. 28; Campbell & Barwick, 1982, fig. 25; Clement et al., 2016) and Dipnorhynchus kurikae (Campbell & Barwick, 2000, fig. 4).

Mesencephalon: (Nieuwenhuys, Hans & Nicholson, 2014) noted that a rise in the brain roof from the optic tectum marks the diencephalon-mesencephalon boundary in lungfish while Northcutt (1986) noted that, in Neoceratodus, the diencephalon and mesencephalon can also be differentiated by a widening observed in dorsal view. A widening and rising on the dorsal surface are pronounced in NHMUK PV P56035 in the region that would be expected to correspond to the mesencephalon-diencephalon. These swellings likely correspond to the optic lobes, which appear unified in ‘C.’ australis in contrast to Dipterus valenciennesi, where paired swellings are observed laterally on the dorsal surface of the endocast (Challands, 2015, fig. 6a). This region could not be segmented in Rhinodipterus kimberleyensis as a result of damage to the material (Clement & Ahlberg, 2014), but Clement et al. (2016, fig. 2a) noted a slight swelling in the mesencephalon region posterior to the pineal-parapineal recess that may represent the region of the optic lobes. Nonetheless, the proposed optic lobes of ‘C.’ australis appear to be slightly enlarged dorsally relative to other Devonian lungfish, though considerably less than contemporary actinopterygians in which the optic lobes occupy the majority of the midbrain (Mimipiscis, Giles & Friedman, 2014, fig. 4; Raynerius, Giles et al., 2015, fig. 3; Giles, Rogers & Friedman, 2016).

In ‘Chirodipterus’ australis the orientation of the brain regions anterior to the mesencephalon is markedly different to that of the mesencephalon and more posterior rhombencephalon. Both specimens of ‘Chirodipterus’ australis exhibit a ventral deflection of the brain regions anterior to the mesencephalon. This is also seen in Dipnorhynchus and Rhinodipterus kimberleyensis whereas in other underformed dipnoan endocasts the brain regions all lie in the same plane. Eusthenopteron also possesses this character and so it is not limited to lungfish, and in Qingmenodus the brain regions are aligned in the same plane but the olfactory tracts and olfactory bulbs are inclined ventrally.

The arrangement of the trigeminal complex and the facial nerve in ‘C.’ australis is identical to that seen in Dipterus valenciennesi, Chirodipterus wildungensis, Dipnorhynchus and Rhinodipterus kimberleyensis (Challands, 2015; Säve-Söderbergh, 1952; Campbell & Barwick, 1982; Campbell & Barwick, 2000; Clement & Ahlberg, 2014). The trigeminal was not described as branching in Rhinodipterus kimberleyensis (Clement & Ahlberg, 2014, fig. 2c), however both separate branches can be seen to be present as in ‘C.’ australis. In Griphognathus whiteii, the anterior branch of the trigeminal, the profundus (V1), exits further to the posterior than in ‘C.’ australis and, currently, seems to be the only dipnoan exhibiting such an arrangement. In Gogodipterus paddyensis and Holodipterus gogoensis the profundus (V1) exits more anteriorly (Miles, 1977, figs. 21 & 22 respectively). Nonetheless, the similarities in the arrangement of the trigeminal complex between the Devonian Dipnoi indicate a relatively conserved trigeminal and facial nerve canal arrangement in the endocast. This is further corroborated by a comparable configuration in Neoceratodus (Northcutt, 1986, fig. 3; Clement et al., 2015). In contrast, the Devonian actinopterygian Mimipiscis toombsi, displays a different configuration for the profundus (V1), mandibularis and maxillaris (V2, V3) and facial (VII) nerves, in which the roots of these nerves are separate (Giles & Friedman, 2014). ‘C.’ australis also differs from the type species C. wildungensis, in which the antero-dorsal portion of the orbital artery sits further laterally away from the nerve canals.

Rhombencephalon: While their function is to drain fluid from the labyrinth, the large endolymphatic ducts topologically occur in the hindbrain and leave the cranial cavity from paired supraotic cavities to join the temporalis fossa in a similar manner to Holodipterus gogoensis, Griphognathus whitei (Miles, 1977) and Dipterus valenciennesi (Challands, 2015). The diameter of the canals for the endolymphatic ducts is, however, much greater in ‘C.’ australis than in these other taxa. The position of the glossopharyngeal nerve (nIX), in ‘C.’ australis is similar to that of Dipterus valenciennesi, Chirodipterus wildungensis (Säve-Söderbergh, 1952), Dipnoryhnchus (sussmilchi—Campbell & Barwick, 1982; kurikae—2000), and Griphognathus whitei (Miles, 1977). Unlike Dipterus valenciennesi however, this canal is not seen to bifurcate. Comparison of this nerve canal with Rhinodipterus kimberleyensis is harder to ascertain as the sacculus of R. kimberleyensis displays no emanating canals, and there is no canal that could correspond with nIX. The canal for the glossopharyngeal nerve nIX is also observed in the same position within the Devonian actinopterygian Mimipiscis toombsi (Giles & Friedman, 2014, fig. 4), implying conservation of the topology of this character between basal actinopterygians and sarcopterygians.

Division into dorsal and ventral branches of the vagus nerve (nX) canal has not been noted in the digital endocast of any dipnoan, although it was described in C. australis by Miles (1977). The division, however, has been noted in actinopterygians (Stensiö, 1927), although the most recent published accounts of Palaeozoic actinopterygian endocasts, while noting the position of a single canal for nX, do not report a division (Giles & Friedman, 2014; Giles et al., 2015). ‘C.’ australis therefore appears unique regarding nX, and future endocast studies will prove interesting in determining whether it will remain a unique character.

The spino-occipital nerves presented here are in accordance with the description by Miles (1977), who also detailed the arrangement in G. whitei. He did, however, note that the spino-occipital nerves vary both intra- and interspecifically in both extant and fossil taxa. Furthermore, it would be impossible to determine that differences in these fine canals were not due to preservation or poor density contrast in CT scans. Suffice to say, ‘C.’ australis possesses a compliment of spino-occipital nerves not unlike that seen in other Devonian lungfish endocasts.

Labyrinth: The sinus superior of ‘C.’ australis clearly extends beyond the dorsal limit of the hindbrain which is considered to be the primitive state for osteichthyans (Giles & Friedman, 2014). This feature is noted amongst the Dipnoi in Chirodipterus wildungensis (Säve-Söderbergh, 1952), Dipterus valenciennesi (Challands, 2015, fig. 9), Dipnorhynchus sussmilchi (Campbell & Barwick, 1982) and,. It is also observed in Youngolepis praecursor (Chang, 1982, fig. 19) and Eusthenopteron foordi (Stensiö, 1963, fig. 50a).

Miles’ (1977) interpretation of the labyrinth of ‘C.’ australis was accurate regarding the general morphology, however he failed to pick up on some of the finer details pertaining to phylogenetically informative characters, which, at the time, was not the aim of the study. No posterior ampulla is depicted in his schematic (Miles, 1977, fig. 48), yet is clearly present in the endocasts of the specimens studies herein. Miles (1977) inaccurate interpretation led subsequent studies to consider ‘C.’ australis as being derived for this character by lacking an enlarged posterior ampulla (Campbell & Barwick, 2000; Friedman, 2007). ‘C.’ australis does in fact possess a posterior ampulla of noteworthy size, as in Dipnorhynchus kurikae (Campbell & Barwick, 2000), Eusthenopteron foordi (Stensiö, 1963, fig. 50a), Youngolepis praecursor (Chang, 1982, fig. 19) and Dipterus valenciennesi (Challands, 2015), and should therefore be coded as primitive for this character. The possession of an enlarged posterior ampulla in ‘C.’ australis further emphasises the retention of several primitive endocranial features in this taxon and potentially supports a more basal phylogenetic position as suggested by recent analyses (Challands, 2015; Clement et al., 2016).

In the uncrushed NHMUK PV P56035, the posterior SCCs have the largest arc of the three SCCs. This contrasts Dipterus valenciennesi and Neoceratodus in which the lateral SCC is the largest in the endocast, but resembles Protopterus dolloi (Challands, 2015, table 1). Arc length is considered to be important in sensing angular acceleration (Popper et al., 2003) and it has been suggested that the lengths of the SCCs relative to each other indicate differences in ecological pressure to detect yaw (sensed by the lateral SCC), pitch (sensed up by the anterior SCC) and roll (sensed by the posterior SCC) to varying degrees (Popper et al., 2003; Challands, 2015).

Unlike the three previously published digital endocasts of fossil dipnoans (Rhinodipterus kimberleyensis, Clement & Ahlberg, 2014; Dipterus valenciennsi, Challands, 2015); Dipnorhynchus sussmilchi Clement et al., 2016, ‘C.’ australis, like the primitive Dipnorhynchus sussmilchi, shows no ventral notch in the sacculolagenar pouch and ‘C.’ australis appears similar to modern lungfish in this regard. The condition in Dipnorhycnhus sussmilchi is hard to discern as the available data analysed by Clement et al. (2016) did not contain the entire sacculolagenar. Campbell & Barwick (1982) reconstructed the sacculolagenar in Dipnorhynchus sussmilchi as being contiguous and if correct would suggest that separation into differentiated saccular and lagenar pouches is the derived condition for this character. Clement & Ahlberg (2014, fig. 4) reviewed the morphology of the labyrinths of eight dipnomorphs, and it appears, currently, that Rhinodipterus kimberleyensis and Dipterus valenciennesi are unique in this trait for Devonian lungfish. Youngolepis praecursor may possess a slight notch in the sacculolagenar, however the vestibular fontanelle obscures the ventral portion of the sacculolagenar making such a claim unsubstantiated.

A notable feature of the endocasts of ‘C.’ australis is the extremely small utricular recess. In this character ‘C.’ australis resembles Lower Devonian dipnoans such as Dipnorhynchus sussmilchi (Clement et al., 2016). In contrast are Dipterus valenciennesi (Challands, 2015) which, although Middle Devonian in age, appears more derived by possessing an enlarged utricular recess, Chirodipterus wildungensis and Rhinodipterus kimberleyensis (Clement & Ahlberg, 2014) which have similarly enlarged utricular recesses. Clement & Ahlberg (2014) first suggested that enlarged utriculi developed throughout the Dipnoi, and with an increasing number of endocasts this is being shown to be the case, though the retention of this primitive character in the Upper Devonian ‘Chirodipterus’ australis requires exploration. The interrelationships of Devonian stem-group lungfish are poorly understood. ‘Chirodipterus’ australis and Dipterus valenciennesi have previously been found to resolve in a polytomy between primitive lungfish (Dipnorhynchus, Uranolophus) and the lungfish crown in the most recent species-level analyses (Clement, 2012; Challands, 2015), while in other analyses relating to neurocranial characters ‘Chirodipterus’ australis is in fact basal to Dipterus valenciennesi (Friedman, 2007). In the phylogenetic analysis here however (which is based on Clement et al., 2016), Dipterus resolves as basal to ‘C.’ australis though with a Bremer support of 0,. Either this is a truly retained primitive character in ‘Chirodipterus’ australis (with independent acquisition of enlarged utricular recesses in Dipterus and derived dipnoans) or ‘C.’ australis exhibits reversion to the primitive state. With the uncertainty of the positioning of ‘C.’ australis and the multiple primitive characters observed in this study, the former hypothesis seems more likely. Furthermore, the necessity for homoplastic expansion in Dipterus on one hand and derived dipnoans on the other (Chirodipterus wildungensis, Rhinodipterus, Neoceratodus and Protopterus; Säve-Söderbergh, 1952; Clement & Ahlberg, 2014; Challands, 2015, table 1), would be rendered unnecessary should ‘C.’ australis settle out as basal to Dipterus valenciennesi. Just how far back within sarcopterygian evolution this trend may extend is not yet known, though it was noted by Miles (1977) that the labyrinth of Eusthenopteron foordi (Stensiö, 1963, fig, 50a) was similar to that of ‘C.’ australis. Unfortunately, the utricular recess was not observed in the endocast of the tetrapodomorph Gogonasus (Holland, 2014), however the onychodont Qingmenodus (Lu et al., 2016, fig. 2b) possesses a small utricular recess not wholly distinct from the semicircular canals. Interestingly, in the most basal sarcopterygian Ptyctolepis (Lu et al., 2017) the utricular recess is largely undifferentiated from the anterior and lateral semicircular canal ampullae compared to the utricular recesses of Devonian lungfish, suggesting that clear differentiation of the utricular recess from the semicircular canals may be an autapomorphy of Dipnoi, with a trend of increasing size of the utricular recesses developing in progressively crownward Dipnoi.

Intraspecific variation in the endocast of ‘Chirodipterus’ australis

To date, endocranial examination of fossil sarcopterygians has taken the format of study on singular, well-preserved specimens, resulting in an inability to even begin to assess the degree of intraspecific variation in endocranial anatomy. Having two skulls of ‘Chirodipterus’ australis presents an opportunity to do just this for the first time in sarcopterygian fossil fish. Given the paucity of exquisitely preserved fossil material available for 3D digital examination, interpretations are susceptible to both under-estimation of intraspecific variation and overstatement of minor differences as phylogenetically informative. A number of differences have manifested between the endocasts of the two specimens of ‘Chirodipterus’ australis examined herein, which warrant discussion.

The cranial nerves of the specimen display very little variation in topology, however nerves IV and VI were not observed in both specimens (lacking in NHMUK PV P56038 and P56035 respectively). Given that these are very small nerve roots, and given the morphological conservatism of the remainder of the cranial nerves, these differences can be attributed to preservation error.

A major difference between the two endocasts is the bifurcation of the olfactory nerves and the angle of their projection from the cranial cavity. In NHMUK PV P56038, bifurcation is observed with the nerves emanating from the cranial cavity horizontally on the same plane as the cavity, while in NHMUK PV P56035 bifurcation is not seen and the nerves project down from the cranial cavity. The reconstruction of Youngolepis praecursor was shown to display a small bifurcation of the right olfactory nerve, though not in the left olfactory nerve (Chang, 1982, fig. 17), so there is not only potential for intraspecific variation in fossil taxa, but within specimens. With regard to the bifurcation of the canals it is likely that both between- and within-individuals, differences in the growth of soft nervous tissue between the brain and nasal sacs would have occurred, and therefore the miniscule degree of ossification required to observe bifurcation in a fossil specimen may have been impacted by the size and structure of the soft tissue. While the downward angle of the olfactory nerve canals in NHMUK PV P56035 is drastic, as mentioned above it is unlikely an artefact of preservation. The olfactory nerve canals of NHMUK PV P56038, while roughly horizontal, do exhibit a marginal downward angle (Table S2), though this specimen underwent some minor dorso-ventral compression during preservation that must be considered. We attribute this difference largely to intraspecific variation, as it is possible that it is due to generalised morphing of the cranium (and therefore cranial nerves) to the deep, short snout of this taxon, however in this instance a minor degree of difference may be attributable to intraspecific variation. It is not possible to truly ascertain whether these differences are due to natural variation or preservational, but given the high quality of preservation of NHMUK PV P56035 and P56038, intraspecific variation is more likely with regard to the olfactory nerve canals.

The anterior cerebral vein presents a difficult intersection of interpretation and the potential for intraspecific variation. Specimen NHMUK PV P56035 displays canals emanating from the olfactory nerve canals that are unambiguously the anterior cerebral veins. NHMUK PV P56038 possesses a large, singular emanation from slightly anterior to the pineal recess, as well as tiny buds on the dorsal surface of the olfactory nerve canals either of which may represent the anterior cerebral vein. The anterior cerebral vein of the tetrapodomorph Gogonasus (Holland, 2014) was recently shown to exhibit a form identical to the former possibility, however no dipnomorph endocast has. In contrast, in all dipnomorphs for which the anterior cerebral vein is known, they take the form of paired canals emanating from the dorsal region of the olfactory nerve canals (with some variation in the exact point along the olfactory nerve canals that they emanate). With this in mind, it is likely that the structure observed anterior to the pineal in NHMUK PV P56038 is an artefact of preservation, and that the small buds on the dorsal surface of the olfactory nerve canals represent the roots of the anterior cerebral vein that have been lost to preservation.

A large structure missing in NHMUK PV P56038 is the notochord, which can be assumed to have been lost due to preservation due to the notochord’s critical functional role. Furthermore, the posterior region of this specimen has sustained damage, with the left labyrinth missing; these differences in the endocasts can be safely attributed to preservation.

NHMUK PV P56035 lacks the canals for the ramus opthalmicus profundus and the ramus palatinus VII in the anterior region of the endocast, while NHMUK PV P56038 is lacking the pituitary vein. These structures serve important neurological and vascular functions and so their absence is unlikely due to natural intraspecific variation, but rather loss of the fine-scale canals during preservation.

Phylogenetic results

The phylogenetic analysis under implicit enumeration produced a single most parsimonious tree of length 36 steps (CI = 0.639, RI = 0.667) with ‘Chirodipterus’ australis resolving basal to Chirodipterus wildungensis and in a polytomy with Dipterus (Fig. 10). This result differs to that of Clement et al. (2016) in which ‘C.’ australis is more derived than Dipterus, and differs from that of Challands & Den Blaauwen (2017) where ‘Chirodipterus’ australis resolved more basally to Dipterus. Parsimony analysis using the ‘Traditional search’ in TNT also produced a single tree of the same length, CI and RI and resolved the same topology as Challands & Den Blaauwen (2017; see Fig. S1).

Figure 10 50% Majority Rule tree of Devonian sarcopterygians.

50% Majority Rule tree of Devonian sarcopterygians derived from cranial endocast chraracters of Clement et al. (2016) using implicit enumeration. The numbers at each node represent Bremer support values, with values of 0 indicating nodes that collapsed in the Strict Consesus tree. Brain regions are colourised following the scheme of Clement & Ahlberg (2014): telencephalon, green; diencephalon, red; mesencephalon, blue; labyrinth, orange; rhombencephalon, yellow.

The Dipnoi are characterised by two unambiguous endocranial characters; character 1 (ventral face of the nasal capsule unossified) and character 11 (notochord does not extend to or beyond the level of n.V). Character 2 (buccohypophyseal canal does not pierce parasphenoid) is found in all lungfish crownward of Dipnorhynchus though is noted to be polymorphic for Dipterus. The presence of a separate foramina for the internal carotid artery and efferent pseudobranchial artery is also characteristic of all lungfish except for Neoceratodus in which the condition is reversed.

Discussion

The labyrinth in Devonian lungfishes

Besides the sacculus and lagena, the utricular recess senses acceleration in the vestibular system and the size of this organ, relative the otoconial mass it contains, is a function of sensitivity to acceleration (Challands, 2015). Miles (1977) depiction of the size and shape of the utricular recess was accurate, with ‘C.’ australis displaying a surprisingly small utriculus for its Upper Devonian age. Extant dipnoans possess vastly enlarged utriculi, with Rhinodipterus kimberleyensis (Clement & Ahlberg, 2014), and Dipterus valenciennesi (Challands, 2015) exhibiting enlargement to a slightly lesser degree. Comparison with Dipterus valenciennesi, Neoceratodus and Protopterus of the surface area of the utricular recess to the sacculolagenar show ‘C.’ australis to have the smallest ratio, further corroborating its unique state. Campbell & Barwick (2000) state that Dipnorhynchus kurikae possesses a large utricular recess, however, a rendering of the anterior region of the vestibular region in Dipnorhynchus sussmilichi by Clement et al. (2016) reveals a distinctively small utricular recess. Until the condition can be unequivocally confirmed for Dipnorhycnhus kurikae, we consider the rendering of Clement et al. (2016) to be the most reliable, and a definite indicator that the condition of the utricular recess in the primitive dipnoan genus Dipnorhycnhus is of the primitive state i.e., small.

Patterns in Devonian dipnoan endocast evolution

The endocast of Chirodipterus appears unusual for the Dipnoi in possessing ventrally-directed olfactory tracts though this character is actually seen in other Devonian lungfish (Dipnorhynchus and Rhinodipterus kimberleyensis) and also Eusthenopteron. Chirodipterus wildungensis does not exhibit this condition lending further morphological support for Chirodipterus actually representing more than one genus. This inclination of the anterior brain regions may be a product of shortening of the cheek region that has occurred in Chirodipterus (Campbell & Barwick, 1982) though Dipnorhynchus and Rhinodipterus kimberleyensis do not possess similar shortening. Indeed, Rhinodipterus kimberleyensis is considered as a long-snouted form. Shortening of the endocranial cavity in dipnoans was noted by Clement et al. (2016) to be convergent with Actinopterygii (Giles et al., 2015) and was attributed to the loss of the intracranial joint, though shortening by ‘folding’ appears to be another way in which the length of the endocranial cavity can be reduced. In this respect, lungfish may have adopted a similar strategy to birds where the brain becomes ‘folded’ to accommodate the increase in size of the cerebrum (Balanoff et al., 2013). The shortening of the cheek region and indeed the short snout in Chirodipterus to produce the unusual ventrally directed olfactory tracts appears to be independent of this character as it is shared with the tetrapodomorph Eusthenopteron which possesses an intercranial joint.

The phylogenetic placement of ‘Chirodipterus’ australis under stepwise addition and TBR (‘Traditional search’ in TNT) is consistent with previous Devonian lungfish phylogenies (Challands & Den Blaauwen, 2017) and demonstrates the relatively basal position of ‘Chirodipterus’ australis. This result is important for two reasons. Firstly, it suggests that several features pertaining to the neurocranium are homoplastic. Dipterus may be bracketed by two species of Chirodipterus that both lack a buccohypophyseal foramen in the parasphenoid and so this character may have evolved independently in both species of Chirodipterus. Secondly separation of ‘Chirodipterus’ australis from Chirodipterus wildungensis provides further indication that Chirodipterus is a polyphyletic genus and warrants the need for a renewed analysis and definition of this group. This is further supported by morphological features of the endocast that are not coded in the character matrix i.e., the ventral deflection of the anterior brain regions present in ‘Chirodipterus’ australis but absent in Chirodipterus wildungensis.

The implicit enumeration parsimony analysis further supports Chirodipterus as an unnatural grouping. Whereas this analysis recovers the same length, CI and RI with ‘Chirodipterus’ australis as the basal sister taxon to Chirodipterus wildungensis, the genus is found to be paraphyletic. The node support (Bremer decay indices) for ‘Chirodipterus’ australis and Dipterus in their respective trees is weak indicating the instability of these two taxa. The implicit enumeration analysis resolving both species of Chirodipterus as sister taxa does not resolve the problems concerning homoplasy mentioned above. Rather different characters (characters 7 and 12) become homoplastic instead of the character pertaining to the buccohypophyseal foramen (character 3).

The definitions of suitable phylogenetically informative characters from endocast data in early osteichthyans are still undergoing refinements as new data comes to light. The present analysis demonstrates that the reasonably high consistency index (low homoplasy index = 0.361) of the single resulting tree indicates a strong phylogenetic signal using all the endocranial characters defined by Clement et al. (2016). However, it is important to note that the penetration of the parasphenoid by the buccohypophyseal canal (character 3) is not clear from the endocast alone—it must be judged in the context of the parasphenoid as well. The same applies to character 6, the presence/absence of a pineal foramen. To test whether or not these characters had an overt effect on grouping the lungfish as a monophyletic unit we conducted a further analysis under the same conditions as stated above but eliminating these characters. The new trees produced under both implicit enumeration and stepwise addition + TBR have exactly the same topology albeit with a slightly lower CI and RI (0.618 and 0.639 respectively). This result provides a good test for the phylogenetic efficacy in primitive osteichthyans of characters derived purely from the cranial endocast without reliance on coevolution between extraneous bones.

Conclusions

This study has shown that the endocast of ‘Chirodipterus’ australis displays a combination of derived and primitive neural characters. More primitive characters are observed than previously thought, such as the enlarged posterior ampulla and small utriculus, which will imply placement of ‘C.’ australis in more basal phylogenetic positions. Differences with the Chirodipterus type species are highlighted, further demonstrating that ‘C.’ australis is unlikely a true a member of the genus Chirodipterus. A number of new phylogenetically informative characters are also defined, namely the configuration of canals surrounding the nasal sac and bifurcation of the olfactory nerve canals, along with proposed revisions of existing character polarities (e.g., sessile and pedunculate olfactory bulbs). Rather than implementing these tentative new characters in a phylogenetic analysis, we have tested the efficacy of currently accepted cranial endocast characters. The consistency of the tree topologies resolved under different analytical conditions and with the removal of characters related to but not exclusively concerning the cranial endocast demonstrates its phylogenetic robustness. It is recommended that efforts focus on basal dipnoans and dipnomorphs (e.g., Uranolophus and Youngolepis) along with derived Upper Devonian taxa (e.g., Holodipterus and Griphognathus) in order to resolve the phylogeny of the Dipnoi.

Institutional Abbreviations

NHMUK Natural History Museum, London, UK

Supplemental Information

Supplemental Information 1 Comments of the character matrix of Clement et al., (2016) and Supplementary Figure 1

Single most parsimonious tree using stepwise addition with 10,000 random addition sequence replicates holding five trees at each step, with tree bisection and reconnection (TBR) enabled. Length = 36, CI = 0.639, RI = 0.667. Values at nodes are for Bremer indices.

Click here for additional data file.

Supplemental Information 2 .nex file of the matrix used in the phylogenetic analysis

Click here for additional data file.

Table S1 Measurements of the brain volumes of Chirodipterus australis.

Measurements of the brain of Chirodipterus australis specimens NHMUK PV P56035 and NHMUK PV P56038. V.Tel.Olf, volume of the telencephalon and olfactory nerve canals; V.Di, volume of the diencephalon; V.Mes, volume of the mesencephalon; V.Rho, volume of the rhombencephalon; V.Lab, volume of the endosseous labyrinth; V.ven.exp.t, volume of the ventral expansion of the telencephalon; V.b.olf, volume of the olfactory bulbs.

Click here for additional data file.

Table S2 Measurements of the brain of Chirodipterus australis.

Measurements of the brain of Chirodipterus australis specimens NHMUK PV P56035 and NHMUK PV P56038. “R” and “L” indicate right and left side of the endocast, respectively. ∠ nI.nI, angle between the olfactory nerve canals; ∠ nI.Tel, downward angle of olfactory nerve canals from telencephalon; na.Tel, distance from the nasal capsule to the telencephalon; na.b.olf, distance from the nasal capsule to the bottom of the olfactory bulbs; D.nI, diameter of the olfactory nerve canals; nII.wid, width of the optic nerve II; nII.dep, depth of the optic nerve II; d.end, width of the endolymphatic ducts; d.end, depth of the endolymphatic ducts.

Click here for additional data file.

Table S3 Measurements of the semicircular canals of Chirodipterus australis.

Measurements of the semicircular canals of Chirodipterus australis specimens NHMUK PV P56035 and NHMUK PV P56038. Larc.(Assc/Pscc/Lscc), length of arc of anterior/posterior/lateral semi-circular canal; Int.C.(Assc/Pscc/Lscc), internal circumference of anterior/posterior/lateral semi-circular canal; Mj.Ax.(Assc/Pscc/Lscc), major axis of anterior/posterior/lateral semi-circular canal fenestra; Mn.Ax.(Assc/Pscc/Lscc), minor axis of anterior/posterior/lateral semi-circular canal fenestra.

*Measured structure was dorso-ventrally compressed.

**Dorso-ventral compression may have altered the position of the anterior ampulla and posterior ampulla.

Click here for additional data file.

Table S4 Measurements of the inner ear of Chirodipterus australis.

Measurements of the endosseous labyrinths of Chirodipterus australis specimens NHMUK PV P56035 and NHMUK PV P56038. The contrast in the left labyrinth was too low to confidently determine divisions between the ampullae, utriculus and sacculus. ∠ AsccPscc/Ascc.Lscc/Pscc.Lscc, angle between anterior and posterior/anterior and lateral/posterior and lateral semi-circular canals; SA.(Ascc/Pscc/Lscc).Amp., surface area of anterior/ posterior/lateral semi-circular canal ampullae; SA.UR, surface area of utricular recess; SA.SL, surface area of sacculolagenar; SA.UR/SA.SL, ratio of surface of utricular recess to surface area of sacculolagenar; V.(Ascc/Pscc/Lscc).Amp, volume of anterior/posterior/lateral semi-circular canal ampullae; V.UR, volume of the utricular recess; V.SL, volume of the sacculolagenar; V.UR/V.SL, ratio of volume of utricular recess to volume of sacculolagenar.

*Measured structure was dorso-ventrally compressed.

Click here for additional data file.

The authors would like to acknowledge the reviewers P Ahlberg and A Clement for their constructive and informative comments that improved the manuscript greatly. We would also like to thank D Sykes and E Bernard for access to and assistance scanning the specimens.

Additional Information and Declarations

Competing Interests

Author Contributions

Data Availability

The authors declare there are no competing interests.

Struan A.C. Henderson performed the experiments, analyzed the data, prepared figures and/or tables, authored or reviewed drafts of the paper, approved the final draft.

Tom J. Challands conceived and designed the experiments, analyzed the data, prepared figures and/or tables, authored or reviewed drafts of the paper, approved the final draft.

The following information was supplied regarding data availability:

Challands, Tom (2018): Segmented endocast of ‘Chirodipterus’ australis BMNH P56035. figshare. Dataset. https://doi.org/10.6084/m9.figshare.6265496.v1.

Challands, Tom (2018): Segmented endocast of ‘Chirodipterus’ australis BMNH P56038. figshare. Dataset. https://doi.org/10.6084/m9.figshare.6265490.v1

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
