# Peer review of "The cranial endocast of the Upper Devonian dipnoan ‘Chirodipterus’ australis"

_PeerJ, doi:10.7717/peerj.5148_

## Round 0.1 · original submission · Major Revisions

Dear Struan,

I have now received two reviews of your paper submitted to PeerJ. The reviewers recommend a number of improvements, which should be addressed before re-submission.

I suggest you colourise your endocast reconstruction so as to make it easier on the reader to distinguish veins/arteries/cranial nerves etc. I would also recommend that you provide a 3D pdf or the stl file of your reconstruction.

Please, together with your unmarked revised manuscript, provide a marked-up copy as well as a document explaining how you have addressed each of the points raised by the reviewers.

Thank you for your attention.

Best regards,
Fabien

·

Basic reporting

• I was unable to find any information in SI regarding the codings for Chirodipterus used in the current analysis. A full character list or matrix is not necessarily required but at a minimum you should include the codings for C. australis.
• The raw scan slices are available via figshare, but it might be nice if the authors could include a simple spinning animation or similar of the segmented endocasts
• I would suggest perhaps moving the tables to SI rather than include then in the main text

Experimental design

No comment

Validity of the findings

No comment

Additional comments

This is a considered and professional contribution to palaeoneurology. The language is (mostly) clear and unambiguous with just a few areas marked on the manuscript requiring further clarification. The introduction and background are thorough and give good context for the article. The figures are high quality and clear, with relevant features clearly labelled and label abbreviations given in the figure captions for ease of interpretation. Some minor grammatical/spelling errors are highlighted in yellow in the attached PDF.

The research question is well defined and fills an obvious gap in our current understanding of lungfish endocasts. The investigation was rigorous and performed to a high technical standard resulting in one of the best quality studies of lungfish endocasts known thus far.

The conclusions are well supported and link clearly to the original research question. This is an important addition to the growing body of work on lungfish cranial endocasts and the early neurobiological evolution of the group.

In addition to those marked on the PDF, some further points for attention:
• It might be nice on one or more of the figures to include lines showing the approximate boundaries between the major brain regions – or include an additional figure showing this
• Remember to italicize taxon names in references
• Supp Info – list Qingmenodus species epithet in tree

·

Basic reporting

There are no serious problems in this area. A number of minor niggles regarding language and references are flagged up under "general comments for the authors"

Experimental design

No problems.

Validity of the findings

No problems.

Additional comments

This manuscript is a valuable contribution to our understanding of Devonian lungfishes. There's nothing fundamentally wrong with it, but having said that there are quite a lot of minor-ish problems that need to be addressed. I have flagged it for 'major revision' because I think it will need to be re-reviewed after revision.

My comments are listed by line number as given in the manuscript.

Line 46: You should explain that you are talking about brain histology here. There's lots of histological data from fossils, but almost all of it is hard-tissue histology.

Lines 47-50: The parentheses around the sentence beginning "Stensiö's inferences..." are really awkward and also unnecessary.

Lines 69-73: You say that Campbell and Barwick 2000 argued that pedunculate olfactory bulbs (and a range of other characters) are primitive for lungfishes, and then say that Clement et al. (2016) have "for the best part" validated this hypothesis, but Clement et al. actually showed that Dipnorhynchus has sessile olfactory bulbs, which, if anything, contradicts the hypothesis rather than validating it.

Lines 73-77: Careful! These are characters of the primitive lungfish Dipnorhynchus, but that does not necessarily equate to the actual characters being primitive for lungfishes.

Line 82: You introduce Rhinodipterus without a citation, making it look like it is described along with Dipterus in Challands 2015.

Lines 84-85: Are these characters also present in modern lungfishes? You imply that they are, but I'm not so sure.

Line 88: There's a space missing between "the" and "posterior".

Line 89: There's a space missing between "ampullae" and "(inflated". The entire sentence spanning lines 88-90 is awkward and clumsy.

Line 169: It is not clear whether "fig. 2" here refers to Fig 2 in Challands 2015 or in the present paper. Also, a guide figure for the measurements would be useful.

Lines 203-204: Two "using" in one sentence is definitely overkill.

Line 237: Here and in many other places we hit a problem that you need to take a strategic grip on. It is perfectly clear from your data that "Chirodipterus" australis cannot possibly belong to the same genus as Chirodipterus wildungensis. Your Figure 9 shows this very clearly: the two endocasts are not merely distinguishable but completely different from each other. It puzzles me why you haven't taken this opportunity to erect a new genus for "C." australis, but in any case you really need to indicate in the text that these two lungfishes don't belong together. How about writing "Chirodipterus" australis throughout?

Lines 242-245: You are missing something important here: Devonian lungfishes as a group are renowned for their high levels of asymmetry, as shown in the dermal bone pattern of the skull. It may be that you are picking up the same signal in the cranial cavities. I suggest you read up on this topic and include a brief discussion here.

Line 247: Strictly speaking, what you are describing is not the nasal capsule (which is a cartilage, eventually fused and co-ossified with the rest of the snout) but the space within the nasal capsule that housed the nasal sac.

Line 261: "carries" should be "continues", I think.

Lines 277-281: The anterior cerebral vein canals you describe from the two specimens are remarkably different; I feel you brush this off rather too lightly. In fact, there's a thematic issue here, where I feel you have made less of your data than you could have done. Almost everything we know about sarcopterygian endocranial cavities comes from single specimens: we have ONE Eusthenopteron, ONE Tungsenia and so forth. This means that it is impossible to determine the degree of individual variation within taxa. Given our human propensity for trying to find patterns, I suspect there's a real danger that we both underestimate the degree of within-taxon variation and over-interpret minor differences between the exemplars we have as phylogenetically important signals. It was thus a very good idea on your part to scan two skulls of "C." australis - but the way you handle the comparison between the two is rather disappointing. I would like to see a systematic overview of similarities and differences between the two specimens, where you discuss in each instance whether the difference is definitely real (as it clearly is in the case of the anterior cerebral vein) or could be due to preservation problems (which should be described: just saying "as being one of preservation", as you do on line 287, doesn't really cut it). This would really advance our understanding of this important topic.

Line 307: I can't see the bifurcation or trace the paths of the internal carotid and pseudobranchial arteries in the figures.

Line 387: Surely the canal is wider than the notochord, not narrower, as this sentence implies?

Lines 387-390: This sentence is difficult to understand.

Lines 433-434: It is difficult to see the hypophyseal recess in the figures because it is partly covered by the jugular vein canal. Why not include an inset figure of this bit of the cranial cavity without the jugular vein?

Line 437: How could a digital endocast possibly have higher resolution than a physical endocast, when it is an image (slightly degraded, like all images) OF a physical endocast??

Lines 447-450: The sentence beginning "While similarly---" is incomprehensible.

Line 461: "to" should be "from".

Lines 464-469: I'm not so sure about this character polarity. Read Ahlberg 1991 (A re-examination of sarcopterygian interrelationships, with special reference to the Porolepiformes. Zool. J.Linn. Soc. 103, 241-287.) and Clement & Ahlberg 2010 (The endocranial anatomy of the early sarcopterygian Powichthys from Spitsbergen, based on CT scanning. In Elliott, D. K., Maisey, J. G., Yu, X. Miao, D. (eds.) Morphology, Phylogeny and Paleobiogeography of Fossil Fishes, 363-377. Verlag Dr Friedrich Pfeil, München.)

Line 478: Porolepiforms and Powichthys, which are probably also members of the lungfish stem group, show a different character state. See references above.

Line 516: For goodness' sake look carefully! Chang's Youngolepis shows a beautiful bifurcation in the right olfactory tract, carefully drawn in her figs 17 and 18.

Lines 534-539: Surely the ventral deflection of the olfactory tracts is simply a reflection of the skull shape - not only short-snouted but with an unusually deep snout? It is not a case of a 'solution' to anything, just overall morphing of the whole cranial package.

Lines 540-548: The argument presented here ignores the fact that a ventral expansion of the telencephalon is present in Dipnorhynchus.

Line 554: In the juvenile Neoceratodus scanned by Clement et al the olfactory bulbs are not obviously pedunculate.

Line 575: What do you mean by the statement that the endocast of Chirodipterus wildungensis is "inaccurate"?

Line 591: "hypophyseal" should be "recess".

Lines 590-606: You should cite Clement & Ahlberg on Powichthys in this section. Also, you are missing the real point here: Youngolepis and less crownward forms have a buccohypophysial foramen; higher lungfishes do not. If you don't retain the foramen there's no point in having a hypophysial recess that reaches down towards the palate.

Lines 609-610: This sentence contradicts itself.

Line 648: whiteii should be in italics.

Lines 660-664: Why do you discuss the endolymphatic ducts here, rather than in connection with the labyrinth?

Lines 716-729: Why do you not discus the size of the utriculus here?

Lines 732-733: "Chirodipterus" australis does not resolve as the "more basal sister taxon" to C. wildungensis! Two taxa are only sister taxa if they form a clade together. "C" australis and C. wildungensis are separate plesions on the dipnoan stem. This is an important piece of terminology that must be used with precision.

Line 741: A closing parenthesis is missing after "V".

Line 763: Why "reduced"? This suggests that a small utriculus is a derived state, which seems very unlikely.

Lines 774-778: I don't believe lungfishes are using any "method" at all to reduce the length of the endocranial cavity. The shortening of the region between nerve VII and the hypophysis is a passive consequence of the loss of the intracranial joint. The downward flexion of the forebrain in "C." australis is a passive consequence of evolving a deep snout.

Lines 779-780: Why does the relatively short notochordal tunnel indicate shortening of the endocranial cavity? It is a complete non sequitur, these are separate structures.

Line 785: Is it OK to cite a paper in review?

Lines 790-792: Couldn't you have carried out a taxonomic revision of Chirodipterus in this paper? Why wait?

REFERENCES: Why are Latin names in the references all given in plain script, not italics?

Challands & Den Blaauwen 2016, Clement et al. 2016b, and Goloboff 2008 are cited in the text but not listed in the references.

Jarvik 1972, Säve-Söderbergh 1952, Stensiö 1927 and 1963, and Thomson & Campbell 1971 are all cite incorrectly. These are all Journal articles and should be cited with volume and page numbers in the usual way. All are cited correctly in Jarvik's (1980) magnum opus Basic Structure and Evolution of Vertebrates if you want to look them up.

FIGURES: You inconsistently use "NHM" and "NHMUK" in the figures and figure legends.

The cranial cavity models would look much better in colour. This would allow you to distinguish veins (which could be shown in blue) and arteries (red) from the cranial cavity and nerve canals. Why not use Romundina (Dupret et al. 2017. The internal cranial anatomy of Romundina stellina Ørvig, 1975 (Vertebrata, Placodermi, Acanthothoraci) and the origin of jawed vertebrates —Anatomical atlas of a primitive gnathostome. PLoS ONE 12(2), e0171241) as a model?

---

## Round 0.2 · Minor Revisions

Dear Struan,

Thank you for the revised version of your MS submitted to PeerJ. I find it much improved. However, I would need you to take into account the (minor) comments made by Reviewer 2 during the second round of reviews before I can accept your work for publication.

Please, together with your unmarked revised manuscript, provide a marked-up copy as well as a document explaining how you have addressed each of the points raised during this second review.

Thank you for your attention.
Best regards,
Fabien

·

Basic reporting

no comment

Experimental design

no comment

Validity of the findings

no comment

Additional comments

On the whole, the manuscript is fine now, and the coloured figures are a major improvement. However, there are still quite a few minor editorial problems that need to be sorted out:

Line 31-34: This is confusing! It is true that the endocast of Chirodipterus wildungensis was the first to be realised (by Säve-Söderbergh in 1952), but the description you give ("not based on a natural cast or 'steinkern'") does not fit: the broken C. wildungensis braincase does contain a steinkern. Furthermore, it is not immediately obvious how you can test the validity of this cranial cavity reconstruction by presenting the cranial cavity of 'Chirodipterus' australis. Why should that be informative about C. wildungensis, especially when you have already put scare quotes around the generic name of 'C' australis to hint that it isn't a true Chirodipterus? The answer is of course that this entire text segment actually refers to Miles's 1977 description of the cranial cavity of 'C' australis! Then the decription makes sense, because the acid-prepared 'C.' australis specimens do not contain steinkerns, and it also makes sense that you can test Miles's reconstruction by presenting a CT-based reconstruction of the same taxon. In other words, by 'correcting' australis to wildungensis in line 32 without changing anything else you have short-cirquited the entire argumentation. Please sort this out.

Line 36-37: You don't describe a description.

Line 48: A monophyletic clade is like a round circle: there's no other kind.

Line 49: The earliest known lungfish, Diabolepis speratus, comes from the Xitun Formation of Yunnan, China, which is late Lochkovian and thus about 410-415 million years old.

Line 96: There should be an "as" before "opposed".

Line 147: There's no such thing as the "crown of Devonian lungfish". The term "crown" has a very specific meaning in phylogenetics: it is a monophyletic group defined by the existence of living members. The lungfish crown group is the clade containing Neoceratodus, Protopterus and Lepidosiren and their last common ancestor.

Line 161-162: On line 154 you have deleted a claim that C. wildungensis and Rhinodipterus kimberleyensis (NOT "kimberlyensis"! This needs to be corrected throughout) have similar cranial cavities, but here the claim persists. Make up your minds!

Line 263: "has been reached" should be "has reached".

Line 703: The hyophyseal what? You can't use "hypophyseal" as a stand-alone noun.

Line 893: "not bifurcation is seen" should be "bifurcation is not seen".

Line 927: "notochords" should be "notochord's"

Figure legends: Figures 1-8 consistently use Chirodipterus australis, not 'Chirodipterus' australis. Figure 9 correctly uses 'C.' australis, but in Figure 10 we have 'Chirodipterus australis'. Sort this out!

---

## Round 0.3 · accepted · Accept

Dear Struan,
Your manuscript has now been accepted for publication.
Both reviewers made an excellent job, which you might want to acknowledge.
Best regards,
Fabien

#